# Measures of Volatility, Crises, Sentiment and the Role of U.S. 'Fear' Index (VIX) on Herding in BRICS (2007–2021)

Hang Zhang and Evangelos Giouvris *

School of Business Management, Royal Holloway, University of London, London TW20 0EX, UK;
hang.zhang.2018@live.rhul.ac.uk
* Correspondence: evangelos.giouvris@rhul.ac.uk

**Abstract:** We look into determinants (volatility, crises, sentiment and the U.S. 'fear' index) of herding using BRICS as our sample. Investors herd selectively to crises and herding is a short-lived phenomenon. Herding was highest during the global financial crisis (only China was affected). There was no herding during the European debt crisis and COVID. With regard to the relationship between volatility and *CSAD* (cross sectional absolute deviation)/herding, a lower *CSAD* (movement in a specific direction) brings about less volatility. However, a high volatility amplifies herding (reduces CSAD), especially in China. Russia and South Africa are unresponsive to volatility levels (low/high) and herding. We also observe volatility heterogeneity. Different volatility measures have different effects on different markets. There is limited evidence to suggest that sentiment (based on principal component) Granger causes herding/CSAD. Herding is a period and market variant and unrelated to crises. The U.S. 'fear' index has a short-lived, limited effect on CSAD/herding (during COVID only) for all countries except China. In addition, Granger causality analysis indicates a two-way relationship between the U.S. 'fear' index and CSAD/herding, unrelated to crises.

**Keywords:** herding; volatility; sentiment; principal component; crises; COVID; U.S. 'fear'; BRICS

**JEL Classification:** G14; G15; G41

## 1. Introduction

Herding behaviour is identified as mimic behaviour which results in phenomena where people tend to follow others' actions rather than making decisions based on their own private or public information (Banerjee 1992). Despite numerous theoretical frameworks (such as Bikhchandani and Sharma 2001) and empirical evidence (such as Chang et al. 2000) which reference herding, there seems to be a lack of cases focusing simultaneously on all BRICS countries in any one piece of research. Firstly, instead of concentrating on institutional investors or individual BRICS countries (Indārs et al. 2019 for Russia and Banerjee and Padhan 2017 for India), this study will assume a holistic outlook and will provide insights on similarities and differences across BRICS. Secondly, the study will concentrate on the relationship between sentiment and herding using principal component analysis. Thirdly it selects the Coronavirus crisis as a sub-period to test the effect of a non-financial crisis on BRICS, which is the first attempt to the best of our knowledge. Fourthly, in order to bridge the gap as far as the relationship (bi-directional: effect of herding on volatility and effect of volatility on herding) between herding and volatility is concerned, this study will utilise different models to measure volatility. We can then compare if different measures of volatility have diverse effects on herding, rather than looking at the effect of a single volatility measure, which is typical in the past literature (such as idiosyncratic volatility in China (Gong and Dai 2017) and implied volatility as a fear indicator (Economou et al. 2018)). Finally, this study will focus on the role of the US on BRICS (including the effects of the US equity market and US sentiment or 'fear' index),

which will expand research in this aspect. The sample period is between 2 July 2007 and 30 September 2021, and the CCK model (Chang et al. 2000) was utilised for analysis.

### 1.1. Motivation

There were several motivations regarding the sample selection. The first motivation is that the rapid development of BRICS in various aspects is reshaping the global economic environment, making them play an increasingly important role in global economic affairs. Their contribution to the global economy increased from 8% in 2001 to 25% in 2019 (EMIS 2019). Simultaneously, more cooperation among BRICS in areas of cybersecurity and trade technology in the last decades has shifted global attention towards emerging economies, for example, more than 40% of global e-commerce transactions can be attributed to China (McKinsey 2017). Such tendencies are making increasing numbers of global investors realise their development potential and inject more capital to pursue investment opportunities in these markets, resulting in more integration between emerging markets and mature markets. However, the low correlation between emerging and developed markets motivates investors to expand investment in emerging markets for the purpose of diversification (Bekaert et al. 2003). Therefore, selecting BRICS as our sample can provide more information for global investors to better understand the financial environment of emerging markets and to make decisions.

A further motivation is that the different characteristics among BRICS can trigger different behaviour. The return distribution of BRICS exhibits strong volatility clustering and a high-risk premium, indicating significantly high volatility (Adu et al. 2015). To date, a large and growing body of research has investigated the relationship between volatility and investors' behaviour, suggesting that a higher degree of volatility is regarded as one of the triggers of herding (Huang et al. 2015; Lakshman et al. 2013). This is one of the hypotheses tested here. A higher volatility reduces *CSAD* or causes more herding (this hypothesis is supported here). The majority of the prior literature focused on the relationship between a single volatility measure and herding (e.g., idiosyncratic volatility and implied volatility). Few studies look into whether different measures of volatility produce different results in this respect for BRICS. Although Blasco et al. (2012) utilised nine different volatility measures in Spain, they studied the effects of herding on volatility and neglected the effects of volatility on herding. After eliminating the day-of-the week and volume effects, they found that an intensified herding level can trigger higher volatility for historical and realised volatility due to "uninformed trading", whereas this is not the case for implied volatility. Thus, in order to bridge this research gap, this study will employ six different volatility measures to test for the presence of the possible different effects of volatility on herding. We also look at the effect of herding on volatility (the reverse relationship). According to the literature (see Topol (1991), Demirer and Kutan (2006), Jlassi and Naoui (2015)), the null hypothesis is that a lower *CSAD* or greater herding increases volatility (this hypothesis is not supported here).

Additionally, a lack of transparency in corporate information disclosure (including governance and financial information) is a perennial problem for BRICS. For example, 24 practices are governed by national law in India, compared to the 52 recommendations of the International Financial Reporting Standards, and seven companies is the median number of corporations disclosing complete governance information (Oliveira et al. 2016). Evidence of a rise in herding tendency as a result of non-transparency (due to high costs for acquiring information and information asymmetries) is evident in Wang and Huang (2018). Moreover, despite the strengthening of integration with global investment markets, a framework of restrictions on foreign investments and investors still exists. For instance, only 2% of the overall equity market and 2% of the bond market was attributed to foreign investment in China in 2019 (Gill 2020). Similarly, foreign direct investment in South Africa was made up of just 1.31% of GDP, less than the average worldwide level of 4.17% in 2019 (World Bank 2020). Surveys such as that conducted by Choudhary et al. (2019) confirmed that foreign investors were inclined to herd towards others' behaviour due to a lack of

information. Therefore, research in BRICS will contribute to making sound investment decisions and constructing relatively diversified portfolios.

### 1.2. Contribution

This study will focus on herding in BRICS countries and compare investors' behaviour in these countries. In previous research, the majority of scholars paid more attention to developed markets such as the United States (Kabir 2018), or single emerging markets such as the Russian equity market (Indārs et al. 2019). Few studies brought all BRICS together in a single sample to compare whether investors behave in a different manner. The following cases can be deemed as good examples which capture the focus of the prior literature and clearly show how this study differs. Demirer and Kutan (2006) found no herding in the Chinese stock market. This finding was in contrast with Tan et al. (2008) who reported herding in both Chinese A-share and B-share stock markets. Moreover, Júnior et al. (2019) reported herding in the Brazilian stock market, controlling for a number of variables such as volatility and dividend yield. Indārs et al. (2019) investigated the Russian stock market and showed that herding is relevant to fundamental factors during the Ukranian crisis. Lakshman et al. (2013) studied the Indian market and found that herding was limited. Ababio and Mwamba (2017) concentrated on the South African market and found herding in the Banking and Real estate sectors, but the timing of herding was different (herding in the banking industry occurred during bear markets, while herding in the real estate industry occurred in rising markets). Collectively, these studies outlined a critical role of herding in investment decisions. However, these studies have just investigated herding in individual markets rather than all of BRICS. A recent study (which has been made known to us at the final stage of this article) and looks into BRICS herding and the effect of volatility on herding (simultaneously for all countries) is that of Mulki and Rizkianto (2020). These are the only common research elements between our study and their study. Samples considered are different and it seems that in the Mulki and Rizkianto (2020) study there is no discussion for structural breaks (if present and how they control for them). The Mulki and Rizkianto (2020) study captures two crises, namely the Asian crisis and the Global financial crisis, while our study captures the Global financial crisis, the European financial crisis and the COVID crisis (controlling for structural breaks). Considering the common research elements of the two studies which allow a comparison of findings, we report that during the Global financial crisis, herding is present only in China, while Mulki and Rizkianto (2020) report herding in all countries except in India and Russia. So, both studies agree on findings regarding India (no herding), Russia (no herding) and China (herding). It is important to keep in mind that in the Mulki and Rizkianto (2020) study, there are no pre-crisis/during-crisis sub-periods to control for structural breaks which may affect the results. The second research element which allows comparison is the effect of volatility on herding. Our study finds that only high volatility causes herding (3 out of 5 countries, for five different volatility measures), while in Mulki and Rizkianto's (2020) study, any level of volatility (using a single measure) causes herding. Evidence presented between BRICS is mixed. This could be a direct result of the use of dummies while we split the sample in quartiles with each quartile capturing a different volatility level using six different volatility measures. Furthermore, we investigate sentiment, VIX and US investor behaviour on herding (as well as Granger causalities) which are not in the objectives of the Mulki and Rizkianto (2020) study; therefore, this is as far as comparisons can go.

Secondly, when it comes to related research regarding the relationship between volatility and herding, the majority of past studies have used a single volatility measurement, such as the GARCH model. For example, Huang et al. (2015) used idiosyncratic volatility to examine the effect of volatility on herding, and Huang and Wang (2017) employed the volatility index to capture fear and sentiment in their analysis. However, this study will use six different measures, as already indicated, to calculate volatility, including conditional volatility, a realised volatility model and historical volatility measures. The aim is to investigate if and how different measures/models of volatility relate to herding, and if herding

is conditional on specific measures of volatility or volatility in general, no matter how it is measured. As already stated above, our hypothesis is that all volatility measures capture the same effect and have a similar impact on herding. We call this volatility measure homogeneity (this hypothesis is not supported here).

Principal component analysis (PCA) will be utilised in this study to capture investors' sentiment, as there is a lack of studies that examine the relationship between investors' sentiment and herding employing PCA. Past research has simply produced a sentiment index using PCA, however, there was no attempt to test its effect on herding behaviour. We believe that this is another novelty. Other studies have made use of other indicators to capture investors mood such as the US implied volatility (VIX index). In this field, Gavriilidis et al. (2016) observed that investors' sentiment could exert influence on herding behaviour using the US sentiment index (VIX index). Chen et al. (2014) used principal component analysis to capture sentiment, but they did not report its relationship to herding. For similar research see Liao et al. (2011), Hudson (2014) and He et al. (2017). In this study, not only do we incorporate PCA into 'capturing' sentiment, but we also examine its relationship with herding for all BRICS, simultaneously, in an attempt to provide a final answer regarding any effects. Our hypothesis is that sentiment (based on PCA) Granger causes herding (partially supported here, there is limited evidence, unrelated to the crises).

Furthermore, three different crisis periods will be introduced into the analysis in order to explore the effects of individual crisis incidents on investors' behaviour. In addition to the global financial crisis, the European debt crisis and the coronavirus pandemic were selected. Our hypothesis is that crises have an impact on herding (not supported here). There are very few articles focusing on the effects of two or more events on herding in BRICS simultaneously. As indicated previously, the most recent study by Mulki and Rizkianto (2020) captures only two crises. The coronavirus pandemic has swept across the globe, causing not only damage to public health, but also leading to the collapse of stock markets in some countries and general economic upheaval. For instance, the US suffered from the fastest decline in its stock market since 1987, evidenced by four successive circuit breakers within several days (Shieber and Crichton 2020). Therefore, introducing and controlling for coronavirus in this study will enable a comparison of the degree of influence that different crises have on investor behaviour. To the best of our knowledge, there is very little research, if any, on the effect of pandemic(s) on herding, especially for BRICS.

Finally, this study will test whether herding is a short- or long-lived phenomenon by introducing different data frequencies, including daily data, weekly data and monthly data. Our hypothesis is that herding is present when daily data is used (fully supported here). In the previous literature, few studies have focused on covering different types and frequencies of data in the same article, allowing for comparison, especially for BRICS. Hence, this study will fill this gap.

To prompt the results, herding, a mimic behaviour accompanied by the suppression of one's own beliefs, can be 'perceived' as a short-term tendency (for example, herding is present in China when employing daily data, but vanishes when employing weekly or a longer time-frequency). This supports our hypothesis above (herding is present when daily data is used). Moreover, there is no herding reported for the other four markets regardless of frequency. Secondly, not all crises have the same effect. The Global Financial Crisis had the greatest effect on herding (during which only China was affected) but the European debt crisis and Coronavirus had zero impact on herding. Our hypothesis that crises have an impact on herding is not supported. Those crisis events are also imported, taking the form of spill-over effects from the US market. Thirdly, more volatile environments are considered as one of the determinants of herding, especially for China. However, at this point, it is worth noting that Russia and South Africa seem to be indifferent to volatility levels. There is no significant herding in any of the two markets for both low/high volatilities. This provides support to our hypothesis that a higher volatility causes more herding (or reduces CSAD), but not for Russia and South Africa. The above findings reported here shed new light on BRICS' behaviour characteristics, which can be conducive to deciding on

an 'investment timing' strategy for global investors. Fourth, there is limited evidence that sentiment based on PCA Granger causes herding/CSAD. It is period and market specific and unrelated to crises. This provides very little support to our hypothesis above. The analysis undertaken here has extended our knowledge of how sentiment can exert influence on herding, which can provide more evidence to understanding BRICS' herding patterns in a more comprehensive way. Finally, spill-overs between the US market and BRICS, as well as between US "fear" or VIX and BRICS is period and market specific and unrelated to crises. For example, VIX affects herding only during COVID 19 for all countries except China. The null hypothesis ((VIX)-fear index affects investor behaviour (herding or CSAD) in BRICS) is partially supported. Specifically, the absence of an effect between US sentiment/VIX and China implies that China is independent when making investment decisions. Therefore, China can be regarded as one of the alternative markets for investors who wish to construct global portfolios or participate in emerging equity markets. China can provide more diversification opportunities due to its low correlation with developed markets. Finally, we observe a two-way Granger causality between VIX and herding (or CSAD), independent of crises which is not what most financial economists thought. Therefore, the null hypothesis that VIX Granger causes herding (or CSAD) is not supported. This study provides a deeper insight into similarities and differences across BRICS and between emerging markets and developed markets. Most importantly, this study will not only be of interest to those who participate in investments in emerging markets but also is relevant to governments that wish to introduce more sophisticated policies and systems. Section 2 is the literature review. Section 3 will introduce the methodology, model(s) and sample(s) used in this study and Section 4 will discuss findings. Section 5 is the conclusion and Section 6 is implications.

## 2. Literature Review

### 2.1. Models

#### 2.1.1. Theoretical Models

Banerjee (1992) was probably the first person to build a theoretical framework to analyse herding. According to Banerjee (1992), herding behaviour tended to occur in uncertain situations in which investors made buying/selling decisions sequentially due to beliefs that other investors possessed superior private information and made better investment decisions. Meanwhile, Welch (1992) agreed with this perception and established a similar "information model" to discuss herding behaviour as well, as did Bikhchandani et al. (1992). Additionally, Lux (1995) attempted to explain the effects and essence of herding behaviour by a new model named the "infection model". Furthermore, instead of a traditional model, Borensztein and Gelos (2003) applied a Monte Carlo simulation to differentiate "theoretical distribution" and "actual distribution", which can further prove the existence of herding.

#### 2.1.2. Statistical/Empirical Models

Christie and Huang (1995) developed a model named "the CH model" to test herding behaviour during stress markets. They claim that herding can be regarded as a kind of irrational behaviour among investors where stock prices may deviate from their equilibrium price level. Christie and Huang (1995) attempted to demonstrate that herding behaviour can lead to low dispersions, especially under stress market conditions. Despite its popularity in the field of herding, there seems to some criticisms and judgements about the model.

In order to overcome the shortcomings of the CH model, Chang et al. (2000) developed a similar but different model to detect herding effects in equity markets. According to Chang et al. (2000), linear relationships cannot always exist in equity markets. They proposed that non-linear relationships between dispersions of individual stock returns and market returns tended to be more common in actual financial markets, especially for some emerging markets. They introduced cross-sectional absolute deviation of returns (CSAD) as a herd indicator, rather than the direct utilisation of deviation levels based on

the guidance of the unconditional asset pricing model. Compared to the CH model, the CCK model is more suitable, due to its advantages.

### 2.2. Volatility and Herding

Over recent decades, most research looking into the relationship between stock performance and volatility has emphasised the important role of volatility in analysing market conditions and making judgements about possible future directions. For example, both Amata et al. (2016) and Becketti and Sellon (1989) agree that there tends to be a relatively close relationship between volatility and the macroeconomic environment; the phenomenon can be explained by the fluctuation of macroeconomic indicators (such as the growth of market volatility with the rise in interest rates, both in the short run and in the long run), which might result in volatility of the stock market. In addition, excessive volatility can trigger changes in the regulatory system and macroeconomic policy (see Roll (1984), Scott (1991), Yadav (2017), Black (1976), Nelson (1991), Campbell and Hentschel (1992) and Schwert (1990)).

Since experiencing excessive market volatility in the 1980s and several large stock market crashes, looking into the interaction of volatility and behaviour was the next step. According to Topol (1991), herding creates and exacerbates volatility, leading to abnormally high transaction volumes and finally resulting in fluctuations in prices. Likewise, Jlassi and Naoui (2015) echoed Demirer and Kutan (2006) by underlining that herding could be regarded as a significant trigger and ingredient of excessive market volatility, which could have a negative effect on the stability of stock markets. According to the literature, the null hypothesis is that herding/CSAD causes volatility (not supported here). On the other hand, when it comes to the effects of volatility on changes in investors' behaviour, a growing body of empirical evidence can testify their causal relationship. Originally, the first systematic study of the relationship between volatility and market participants' behaviour was reported by Friedman (1953). Friedman stressed that volatility would make investors become more irrational and change their strategies, such as buying high and selling low, destabilise the market, make prices deviate from their fundamental value and increase market inefficiency. Moreover, Choudhary et al. (2019) provided evidence to confirm that market fluctuations can be regarded as a reflection of market uncertainty and inadequate confidence, thus playing a leading role in changes in investment strategies and behaviour convergence. Furthermore, there is a large number of empirical and theoretical studies for developed markets (Blasco et al. 2012; Kremer and Nautz 2013; Ouarda et al. 2013, etc.) and emerging markets (Alemanni and Ornelas 2006; Balcilar et al. 2013; Guney et al. 2017, etc.). According to the literature, the null hypothesis is that a higher volatility causes more herding OR reduces *CSAD* (supported here).

### 2.3. Principal Component, Sentiment and Herding

In addition to the importance of volatility, during the past 40 years, much more information has become available on studies about how investors' sentiment or emotions exert great influence on investors' behaviour and strategies. As Simon and Wiggins (2001) stated, sentiment could be defined as deviations between forecasted stock returns and actual returns, as well as attitudes towards future directions of the market. Since the 1980s, a set of studies have worked on the issue of whether investors' sentiment could induce mispricing by introducing volatility and bubble events (for example, see Shiller (1981); Poterba and Summers (1988)). Similarly, a large volume of published studies has shown that there is a strong correlation between sentiment and the equity market. This is captured by a positive sentiment which is associated with high demand for specific stocks, without consideration for fundamental factors (Arkes et al. 1988; De Long et al. 1990; Wright and Bower 1992). As a result, this phenomenon can result in an overreaction towards the market and an increase in speculative or irrational behaviour by changes in ability and willingness to take risks, especially within a short horizon (Eichengreen and Mody 1998;

Baker and Wurgler 2007). Our hypothesis is that sentiment Granger causes herding/CSAD (partially supported here).

How to measure sentiment, an abstract and subject variable, has become an essential issue to be solved before testing the relationship between sentiment and behaviour. There are several strands in the literature: (1) models based on investor types (De Long et al. 1990; Hong and Stein 1999), (2) models based on cognitive bias or asymmetric information (Barberis et al. 1998; Brown and Cliff 2004); (3) utilisation of stock or option market indicators (e.g., liquidity, implied volatility and volume) as sentiment signals (Baker and Stein 2004; Whaley 2000; Scheinkman and Xiong 2003); and (4) discount of closed-end funds as a reflection of sentiment (Zweig 1973; Lee et al. 1991). In recent years, there has been an increasing amount of literature measuring the sentiment index by a new method named "principal component analysis". In order to reduce dimensionality and the number of noisy variables in the construction process of the sentiment index, the principal component is able to capture the degree of effect of every related indicator on the sentiment index using a linear regression model, which can establish a stable index to reflect investors' mood (Alexander and Dimitriu 2003; Brown and Cliff 2004). Evidence shows that the sentiment index using the principal component method has a stronger predicting power than traditional sentiment measurement methods and can be deemed as a better way to examine the effects of various indicators of investors' emotions, whether in developed or emerging markets (Chong et al. 2014; He et al. 2017).

Importantly, there has been considerable evidence supporting a strong relationship between investors' sentiment and herding (Philippas et al. 2013; Chiang et al. 2013). They identified that herding behaviour could be motivated by the investors' mood, especially for fear and negative emotions; the effect could not be only limited within the USA, but also spread to other markets. That is to say, the emotion of US investors could exert influence on investors' behaviour and induce herd in other markets. The above findings are consistent with the research results of Hwang and Salmom (2006), who asserted that there tended to be a negative relationship between the sentiment index and herding indicator and that the phenomenon could be pronounced in a bull market.

### 2.4. The Role of the US

As far as the importance of the US is concerned, there has been a great deal of literature regarding whether the US can be regarded as a 'weather vane' for financial markets or not, and how it can influence investors' behaviour and strategies in other countries or regions. Chiang and Zheng (2010) provided comprehensive and new evidence on this issue using a sample of 18 worldwide markets. According to their investigations, research results using the *CSAD* model supporting those markets tended to herd towards the US, and this tendency appeared to be intensified during crises over the span of about 21 years. That is to say, a crisis can be easily be spread to other markets, driving a convergence of behaviours in the majority of global markets. Moreover, in support of the hypothesis about the leading role of the US in global financial markets, a similar conclusion that a contagion effect existed between the US market and emerging markets was reported in the research of Luo and Schinckus (2015), who opined that the US market could exert great influence on the behaviour of Chinese investors. Nevertheless, it was surprisingly found that there seemed to be a relatively low probability of spill-over effects between the US markets and eight African equity markets. In other words, behaviour in these African markets was motivated only by domestic factors (Masson and Pattillo 2005). Our hypothesis is that US investor behaviour affects investor behaviour (herding) in other countries (partially supported, country specific and unrelated to crises).

### 2.5. BRICS Research

As indicated in the introduction, research in BRICS concentrates mainly on specific countries, which makes a direct comparison between BRICS difficult. Examples of studies that concentrate on individual countries are given below. Zhu et al. (2020) focused on

institutional herding in the Chinese A-share, indicating significant herding on the buy side in the manufacturing and construction sectors. Ju (2019) concentrates both on the A-share and B-share market(s) and shows that herding is present in China but is more pronounced in downward markets irrespective of A or B share classification. With regard to the Brazilian market, Júnior et al. (2019) suggested that the volatility index cannot explain herding, but crises can exert influence on the degree of herding, as indicated by the increase of the herding level between 2009 and the middle of 2016 following the outbreak of the global financial crisis. Similarly, herding is stronger in decreasing markets in Russia and South Africa (see Indārs et al. 2019; Sardjoe 2012; Seetharam and Britten 2013). In addition, non-fundamental factors appear to trigger herding in Russia (Indārs et al. 2019). With regard to India, a significant relationship between large-cap stocks and herding is present in the study of Chauhan et al. (2020), but there is no industry herding according to Ganesh et al. (2016). Conversely in South Africa, there is evidence of herding in the banking sector in bear markets and evidence of herding in the real estate industry in bull markets (Ababio and Mwamba 2017). The only study (which has been made know to us at the final stage of this article) that looks into herding and the effect of volatility on herding for all BRICS simultaneously is that of Mulki and Rizkianto (2020). Both studies agree on findings regarding India (no herding), Russia (no herding) and China (herding), but there is no common ground as far as the effect of volatility on herding is concerned. The research objectives of the two studies are different as well as the sample period. Furthermore, we investigate sentiment based on principal components, VIX and US investor behaviour on herding (as well as possible Granger causalities).

To reiterate and summarise, our research hypotheses are as follows:

**Hypothesis 1 (H1).** *Crises have an impact on/cause herding in BRICS (not supported, evidence in Table 4).*

**Hypothesis 2 (H2).** *Herding is a short-lived phenomenon/present only when daily data is used (supported based on China only, evidence in Table 4).*

**Hypothesis 3 (H3).** *Greater herding/Low* CSAD *causes an increase in volatility (not supported, evidence in Table 7).*

**Hypothesis 4 (H4).** *Higher volatility causes more herding/reduces* CSAD *(supported, evidence in Table 8, Panel D).*

**Hypothesis 5 (H5).** *All volatility measures capture the same effect and have a similar impact (volatility measure homogeneity) on herding (not supported, evidence in Table 8).*

**Hypothesis 6 (H6).** *Sentiment/SIX (based on Principal Component Analysis and country specific indicators) Granger causes herding/CSAD (partially supported, limited evidence, unrelated to crises, evidence in Table 12).*

**Hypothesis 7 (H7).** *US investor behaviour affects investor behaviour in BRICS (partially supported, country specific and unrelated to crises, evidence in Table 14).*

**Hypothesis 8 (H8).** *US (VIX)-fear index affects investor behaviour (CSAD/herding) in BRICS (partially supported, effect observed only during COVID-19 for all countries except China, evidence in Table 16).*

**Hypothesis 9 (H9).** *US (VIX)-fear index Granger causes CSAD/herding in BRICS (not supported. Two-way relationship is present, independent of crises, evidence in Table 17).*

## 3. Methodology and Data

### 3.1. Data and Sample(s)

The purpose of this research is to investigate whether emerging markets present herding behaviour and respond differently to different market states or periods. A further aim is to compare differences in investors' behaviour when they are placed in the same market conditions or met with the same events. The sample is comprised of: Brazilian IBOVESPA, Russian MOEX index, Chinese CSI 300, Indian S&P BSE SENSEX and South African FTSE/JSE Africa All Share Index. All data is from DataStream and Bloomberg.

The whole sample period covers about 14 years, from 2 July 2007 to 30 September 2021. Apart from the complete period to be analysed, the sample is also broken into smaller periods to consider a number of events such as: the "global financial crisis"; the "European debt crisis" and the "coronavirus crisis", respectively. The "global financial crisis" event is between 15 September 2008 and 31 March 2009. This is based on Lin et al. (2013) and Aït-Sahalia et al. (2012). Moreover, according to Dos Dos Santos and Lagoa (2017), the European sovereign debt crisis is between 1 April 2010 and 31 January 2012. According to the World Health Organization (2020), the first suspected case of coronavirus was reported on 31 December 2019, therefore, the sub-period for coronavirus disease is between 31 December 2019 and 30 September 2021. These sub-periods were selected in this study to determine whether investors' behaviour changed with sudden changes in market conditions, especially in a crisis environment.

In addition to daily data, weekly and monthly data were utilised. Syriopoulos and Bakos (2019) reported herding using monthly data in global shipping equity portfolios, which was similar to Hsieh et al. (2011) undertaking research in the Asian mutual fund market. Sias (2004) and Dasgupta et al. (2011) used quarterly data in their research. Christie and Huang (1995) used both daily and monthly data to test for herding. Caporale et al. (2008) and Alhashim (2018) selected daily, weekly and monthly data to examine herding. The purpose was to compare whether different frequencies of time periods can influence the results. In other words, this analysis can help us to understand whether herding is a short-lived or long-time phenomenon.

### 3.2. Basic Model

The CCK model put forward by Chang et al. (2000) is presented below:

$$CSAD_t = \frac{1}{N} \sum_{i=1}^{N} |R_{i,t} - R_{m,t}| \tag{1}$$

$$CSAD_t = a_1 + a_2 |R_{m,t}| + a_3 (R_{m,t})^2 + e_t \tag{2}$$

where (1) $R_{i,t}$ = return on stock *i* at time *t*; (2) $R_{mt}$ = market return at time *t*; (3) $N$ = the number of sample stocks; (4) $CSAD$ = cross-sectional absolute deviation of returns.

If herding exists, the coefficient $\alpha_3$ will be significant and negative. Moreover, returns were calculated using the following formula: $R = \log\left(\frac{Price_t}{Price_{t-1}}\right)$.

In addition, investors' behaviour in different market stress periods was taken into account. Therefore, the modified model to evaluate the effects of a crisis event on herding is shown as:

$$CSAD_t^{Global/European/Covid\ crisis}$$
$$= a_1 + a_2 |R_{m,t}|^{Global/European/Covid\ crisis} + a_3 (R_{m,t})^{2\ Global/European/Covid\ crisis} + e_t \tag{3}$$

### 3.3. Volatility Models

In this study, the effect of volatility on herding is examined. Most importantly, various types of volatility measures are utilised in the analysis. The aim is to compare whether different measures of volatility can exert the same influence on investors' behaviour.

3.3.1. Forecasting Volatility

Conditional Volatility (GARCH)

ARCH and GARCH, are "stochastic volatility models" and will be used to detect the effects of conditional volatility on herding. A GARCH (1, 1) model is presented below:

$$R_{mt} = \alpha + \beta R_{m(t-1)} + \varepsilon_t \tag{4}$$

$$\sigma_{GARCH(t)} = \sqrt{\alpha + \beta \sigma_{Garch\ (t-1)}^2 + \delta \varepsilon_{t-1}^2 + \eta} \tag{5}$$

where (1) $R_{m(t-1)}$ = first-order lagged variable of $R_{mt}$; (2) $\sigma_{GARCH\ (t-1)}^2$ = first-order lagged variable of $\sigma_{GARCH}^2$; (5) $\varepsilon_t$ = residual term at $t$; (6) $\varepsilon_{t-1}^2$ = square of first-order lagged variable for $\varepsilon_t$; (7) $\sigma_{GARCH}$ = conditional volatility. We use the square root of $\sigma$ to obtain the standard deviation for our calculations.

Exponentially Weighted Moving Average Volatility (EWMA)

In order to address the issue of "Ghost Features" of historical volatility measurements ("Ghost Features" is a term used to capture the presence of extreme events or anomalous data points that can exert influence on volatility forecasting and "severely bias the volatility and correlation forecasts upward", resulting in the distortion of results (Alexander 2008)), J. P. Morgan/Retuers (1996) introduced a method named EWMA. The major differences between EWMA and the historical volatility calculation are that EWMA does not only rely on the decay factor to decide on the weight of past variance, but also attaches higher weights in recent observations, rather than placing equal weight on each observation (Alexander 1998). Moreover, J. P. Morgan/Retuers (1996) outlined that EWMA is a relatively more satisfactory method to predict volatility due to external shocks incorporated into the model and the assumption of conditional distributed returns. Therefore, the 0.94 λ suggested by J. P. Morgan/Retuers (1996) is utilised in this model:

$$\sigma_{EWMA(t)} = \sqrt{\lambda\ \sigma_{EWMA(t-1)}^2 + (1-\lambda)R_{mt}^2} \tag{6}$$

where (1) $\sigma_{EWMA(t)}$ = *EWMA* volatility at time $t$; (2) $\sigma_{EWMA(t-1)}$ = first-ordered lagged volatility; (3) Initial Volatility ($\sigma_{EWMA(0)}$) = the initial return squared.

3.3.2. Intraday Extreme Points Volatility

Parkinson (1980) put forward an extreme value method to estimate volatility of security markets based on returns. In comparison to the calculation of returns based on opening or closing prices, which is the standard approach, Parkinson's (1980) method has proved to be more effort-intensive due to the inclusion of the highest and lowest prices in the model. The model is presented below:

$$\sigma_{P(t)} = \sqrt{\frac{1}{4\sqrt{\ln 2}} \times \frac{1}{n} \sum_{t=1}^{n} P_t^2} \tag{7}$$

Additionally, Garman and Klass (1980) extended Parkinson's (1980) method. They established a structural model to capture volatility, based on different price points within one trading day. In other words, they incorporated extreme values, and opening and closing prices into their calculations. The empirical model is presented below:

$$\sigma_{GK(t)} = \sqrt{\frac{1}{n} \sum_{t=1}^{n} \left[ \frac{1}{2} P_t^2 - (2ln2 - 1)Q_t^2 \right]} \tag{8}$$

where (1) $\sigma_P$ and $\sigma_{GK}$ = volatility of market index at time $t$ based on Parkinson (1980) and Garman and Klass (1980), respectively; (2) $P_t = ln\frac{highest\ point\ at\ time\ t}{lowest\ point\ at\ time\ t}$; (3) $Q_t = ln\frac{Closing\ point\ at\ time\ t}{Opening\ point\ at\ time\ t}$.

### 3.3.3. Equally Weighted Moving Average and Capitalisation (or Value) Weighted Moving Average

First, we present an equally-weighted moving average (*MA*) volatility model based on the past 20 days. The model is presented below:

$$\sigma_{MA\ (t)} = \sqrt{\sum_{t=1}^{20}\left(\frac{1}{20} \times R_{m\ (t-1)}^2\right)} \tag{9}$$

where (1) $\frac{1}{20}$ = daily weight of past 20 days; (2) $R_{m\ (t-1)}$ = market return at $t-1$; (3) $\sigma_{MA\ (t)}$ = volatility at time $t$ based on an equally-weighted moving average.

The second method to measure historical volatility is the capitalisation weighted (*CW*) method. The uniqueness of this method is that it will take fluctuations of individual sample stocks into account, adjusted by their capitalisation before calculation of the market historical volatility. This method is presented below:

$$\sigma_{ICW\ (t)} = \sqrt{\frac{\sum_{t=1}^{T}\left(R_{it} - \overline{R}\right)^2}{T}} \tag{10}$$

$$\sigma_{CW\ (t)} = \sum_{i=1}^{n}\frac{Capitalisation\ i}{total\ \text{capitalisation}}\sigma_{ICW\ (t)} \tag{11}$$

where (1) $\sigma_{ICW\ (t)}$ and $\sigma_{CW\ (t)}$ = volatility of individual sample stocks and market volatility at time $t$, respectively; (2) $T$ = the number of active trading days within one month; (3) $n$ = the number of active sample stocks for every market; (4) weight $t$ = the market capitalisation of individual stocks $i$; (5) *total capitalisation* = the sum of market capitalisation for all active sample stocks for every market stock on a monthly basis; (6) $\overline{R}$ = average of rate of return for individual stock $i$ during one month.

### 3.3.4. Herding and Volatility Models

There is strong evidence of the existence of 'day of the week' effect on stock returns and volatility, which has been researched and proved in prior literature, such as "Monday effect" (Berument and Kiymaz 2001). Simultaneously, a dynamic relationship exists between volatility and current or lagged volume (Wang and Huang 2012). In order to eliminate the effect of the day-of-the-week and volume, we present the model below:

$$\sigma_{it} = \alpha + \beta M_t + \gamma V_t + \eta_t \tag{12}$$

where (1) $\sigma_{it}$ = volatility captured by the above-mentioned methods ($\sigma_{GARCH(t)}$, $\sigma_{EWMA(t)}$, $\sigma_{P(t)}$, $\sigma_{GK(t)}$ and $\sigma_{MA(t)}$, except $\sigma_{CW\ (t)}$); (2) $M_t$ = dummy variable (the value is equal to 1 when it is Monday, otherwise = 0); (3) $V_t$ = trading volume at time $t$ (for ease of calculation, the volume will be multiplied by $10^{-10}$); (4) $\eta_t$ = new volatility value obtained for the various volatility measures after eliminating the effect of day-of-the-week and volume.

Since the $\sigma_{CW\ (t)}$ is based on a monthly basis, the day-of-the-week effect cannot be considered. Only the volume effect can be taken into account (to clarify this is for $\sigma_{CW(t)}$ only):

$$\sigma_{CW(t)} = \alpha + \gamma V_t + \eta_t \tag{13}$$

The $\mid\eta_t\mid$ is the new volatility value that will be utilised in our analysis of herding. After calculating various types of volatility, the next step is to consider the relationship

between herding intensity and volatility. By combining the CCK model (Chang et al. 2000) with historical volatility, we obtain the following regression model:

$$\eta_t = \alpha + \beta CSAD_t + \varepsilon_t \tag{14}$$

where $\eta_t$ = true volatility calculated by the previous methods and denoted as $\eta_{GARCH(t)}$, $\eta_{EWMA(t)}$, $\eta_{P(t)}$, $\eta_{GK(t)}$, $\eta_{MA(t)}$, and $\eta_{CW(t)}$.

The above model will help detect the effect of herding on volatility (if any). If the coefficient of $CSAD_t$ is significant, then there is an effect.

In turn, we will examine whether different levels of volatility can exert any influence on herding. This is achieved by splitting the sample in quartiles according to volatility observed (from lowest to highest). The standard regression model will be used as shown below, but the whole process will be repeated 4 times, once for each quartile.

$$CSAD_{i,t} = a_1 + a_2|R_{m,t}| + a_3(R_{m,t})^2 + e_t \tag{15}$$

where the presence of herd behaviour is confirmed if $a_3$ is significantly negative.

### 3.4. Sentiment and Herding

Principal component analysis (PCA) is a statistical method which is used to draw common elements from related variables. This study utilised PCA to construct a 'sentiment index'. Detailed indicators are shown in Table 8. The relevant model is presented below:

$$SIX = \alpha + \beta_1 TURN(t) + \beta_2 PE(t) + \beta_3 GCPI(t) + \beta_4 GIP(t) + \beta_5 GM2(t) + \beta_6 GER(t) \tag{16}$$

where (1) $TURN(t)$ = turnover ratio; (2) $PE(t)$ = market price-earnings ratio; (3) $GCPI(t)$ = growth rate of $CPI$; (4) $GIP(t)$ = growth rate of industrial production; (5) $GM2(t)$ = change in monthly supply of $M2$; (6) $GER(t)$ = change in exchange rate of domestic currency relative to the US dollar.

As indicated by Hudson (2014), a high sentiment index tends to drive herding, but the impact degree seems to be determined by different market stages or conditions. Conversely, Vieira and Pereira (2015) argued that a negative relationship is present between the sentiment index and herding effects due to the assumption that people are more likely to follow their own beliefs and follow independent strategies. Therefore, in order to shed light on the relationship between the sentiment index and herding, we run Granger causality tests. Causality can be traced back to a seminal paper by Granger (1969), thus the term 'Granger causality'.

### 3.5. The Role of the US
#### 3.5.1. Basic Model

Finally, this study will examine the effects of the US on investors' behaviour outside the US. Hattori et al. (2018) showed that there are spillover effects from the US to other financial markets (especially emerging markets) and those effects seemed to be long-lived. Similarly, Liu and Pan (1997) showed that there were cross-country effects between the US and some Asian markets, as far as volatility is concerned. With regard to herding effects, Lee (2006) and Galariotis et al. (2015) showed that the US played a significant role in explaining non-US investors' behaviour. Investors from other markets tend to mimic US investors' strategies and consider the US market's possible future directions in the process of making their own investment decisions. Therefore, due to the important role of the US in global financial markets, this study investigated the effect(s) of the US on BRICS. The empirical model is as follows:

$$CSAD_{i,t} = a_1 + a_2|R_{m,t}| + a_3(R_{m,t})^2 + \alpha_4 CSAD_{us,t} + a_5(R_{us,t})^2 + e_t \tag{17}$$

where (1) $CSAD_{i,t}$ = cross-sectional absolute deviation of returns for *BRICS* markets, respectively, at time $t$; (2) $R_{m,t}$ = market return of BRICS markets, respectively, at time $t$;

(3) $R_{m,t}{}^2$ = square of market return for BRICS markets, respectively, at time $t$; (4) $CSAD_{us,t}$ = $CSAD$ of US at time $t$; (5) $R_{us,t}{}^2$ = square of market return for US market at time $t$).

It is expected that $a_5$ will be significant and negative if spill-over effects exist between the US and the BRICS. The S&P 500 is used to capture the US market index. Simultaneously, the effects of different time frequencies (high-frequency data relative to low-frequency data) were also considered.

Bathia et al. (2016) and Economou et al. (2018) acknowledged that US investors' emotions had the ability to influence global stock markets. Therefore, in this study we added sentiment as one of the variables that can affect herding and modify the model as follows:

$$CSAD_{i,t} = a_1 + a_2|R_{m,t}| + a_3(R_{m,t})^2 + a_4 VIX_{US,t} + e_t \tag{18}$$

where (1) $CSAD_{i,t}$ = cross-sectional absolute deviation of returns for BRICS markets, respectively, at time $t$; (2) $R_{m,t}$ = market return of BRICS markets, respectively, at time $t$; (3) $R_{m,t}{}^2$ = square of market return for BRICS markets, respectively, at time $t$; (4) $VIX_{us,t}$ = log return of CBOE implied volatility index ($VIX$).

The CBOE implied volatility index used here was established in 1993, and it can be regarded as an indicator of sentiment. It captures investors' fear and uncertainty towards the market (Philippas et al. 2013; Whaley 2000). A significantly negative $a_4$ indicates that "fear" can trigger herding behaviour. Additionally, a crisis period and non-crisis period was utilised in the analysis in order to test the effect of fear under different circumstances.

### 3.5.2. Granger Causality Test

Although the regression models mentioned before were used to test the relationship between herding in BRICS and the US, these models just test for the presence of "mere" correlations between US performance and/or attitude and the BRICS' behaviour. In other words, regression models seem not to be effective to examine whether there is a causal relationship or not. Therefore, in order to address this issue, the Granger causality test will be employed.

### 4. Discussion and Findings

### *4.1. Time-Frequency Effect and Event Effect for Herding*

Table 1 presents results of the Augmented Dickey–Fuller (ADF) used to test for stationarity. All variables are stationary. Table 2 provides descriptive statistics of $CSAD$ and $R_{mt}$. In Russia, the median is lower for both $CSAD$ and $R_{mt}$ while in India, it is relatively higher. Simultaneously, Russia presented the greatest variation and South Africa reported the lowest volatility in $R_{mt}$. Figure 1 shows that crisis events tend to have an impact on $CSAD$ and market returns, especially during the global financial crisis and the COVID-19 disease crisis, as indicated by the shaded areas. Specifically, in Figure 1, the first shaded area captures the Global Financial Crisis. The second shaded area captures the European Debt Crisis and the third captures the COVID-19 Crisis period. As indicated by Calomiris et al. (2012), emerging markets seem to be more sensitive to crisis shocks compared to developed markets. Additionally, Figure 2 proves the nonlinear correlation between $CSAD$ and market returns, confirming the CCK model proposed by Chang et al. (2000) is suitable for this study. In other words, Figure 2 shows data clustering rather than a linear decrease or increase (e.g., line of a 45-degree slope).

Before discussing herding effect(s) in our sample markets, Chow (1960) tests are conducted to determine if there are structural brakes. According to Chow (1960), the null hypothesis is "no breaks at specified points". If the p-value of the F-statistic is less than 0.05, then the null hypothesis is rejected. The three sub-periods under examination are also displayed in Table 3. This includes the period between 2 July 2007 and 31 March 2009, the period between 1 April 2009 and 31 January 2012 and the period between 1 February 2012 and 30 September 2021, respectively. It is worth noting that every sub-period covers one crisis period, which are the Global Financial Crisis (15 September 2008–31 March 2009), European Debt Crisis (1 April 2010–31 January 2012) and COVID-19 Crisis (31 December

2019–30 September 2021), respectively. It can be seen from Table 3 that most markets and sub-periods show a significant F-statistic, except some markets, on a monthly and/or weekly basis. As a whole, the null hypothesis is rejected so there are structural breaks. Splitting the whole sample (2 July 2007–30 September 2021) into smaller samples/periods is justified.

**Table 1.** ADF Stationary Test: Herding Indicator, Market Return and Volatilities.

| Variables | Brazil | China | India | Russia | South Africa | US |
|---|---|---|---|---|---|---|
| **CSAD** | −7.98 | −7.81 | −8.67 | −4.77 | −5.80 | −6.77 |
| (*p* value) | (0.00) | (0.00) | (0.00) | (0.00) | (0.00) | (0.00) |
| $R_{mt}$ | −63.38 | −57.39 | −56.87 | −59.02 | −59.25 | −69.28 |
| (*p* value) | (0.00) | (0.00) | (0.00) | (0.00) | (0.00) | (0.00) |
| $|R_{mt}|$ | −4.51 | −3.72 | −4.48 | −3.22 | −3.79 | −4.67 |
| (*p* value) | (0.00) | (0.00) | (0.00) | (0.00) | (0.00) | (0.00) |
| $R_{mt}^2$ | −7.70 | −4.81 | −8.42 | −5.51 | −3.64 | −8.15 |
| (*p* value) | (0.00) | (0.00) | (0.00) | (0.00) | (0.00) | (0.00) |
| *VIX* **Index** | | | | | | −65.88 |
| (*p* value) | | | | | | (0.00) |
| $\eta_{GARCH(t)}$ | −6.51 | −5.82 | −7.40 | −4.40 | −5.95 | |
| (*p* value) | (0.00) | (0.00) | (0.00) | (0.00) | (0.00) | |
| $\eta_{EWMA(t)}$ | −5.93 | −5.17 | −6.37 | −3.13 | −4.54 | |
| (*p* value) | (0.00) | (0.00) | (0.00) | (0.00) | (0.00) | |
| $\eta_{P(t)}$ | −10.78 | −4.42 | −11.05 | −3.44 | −4.64 | |
| (*p* value) | (0.00) | (0.00) | (0.00) | (0.00) | (0.00) | |
| $\eta_{GK(t)}$ | −10.50 | −4.67 | −5.86 | −3.65 | −5.04 | |
| (*p* value) | (0.00) | (0.00) | (0.00) | (0.00) | (0.00) | |
| $\eta_{MA(t)}$ | −5.57 | −4.96 | −8.16 | −4.06 | −8.10 | |
| (*p* value) | (0.00) | (0.00) | (0.00) | (0.00) | (0.00) | |
| $\eta_{CW(t)}$ | −5.13 | −6.01 | −8.44 | −3.67 | −5.48 | |
| (*p* value) | (0.00) | (0.00) | (0.00) | (0.00) | (0.00) | |
| **Result** | Stationarity | Stationarity | Stationarity | Stationarity | Stationarity | Stationarity |

**Table 2.** Descriptive Statistics: Herding and Market Return Indicators.

| Variables | | | Brazil | China | India | Russia | South Africa |
|---|---|---|---|---|---|---|---|
| | | Mean | 0.6973 | 0.6562 | 0.8262 | 0.5949 | 0.6847 |
| | | Median | 0.6410 | 0.5978 | 0.7668 | 0.5092 | 0.6207 |
| **CSAD** | | Maximum | 2.7927 | 3.2570 | 3.8298 | 5.5714 | 4.0678 |
| | | Minimum | 0.3086 | 0.2144 | 0.3959 | 0.0038 | 0.2530 |
| | | Standard Deviation | 0.2414 | 0.2716 | 0.2531 | 0.3386 | 0.2819 |
| | | Mean | 0.0088 | 0.0032 | 0.0172 | 0.0109 | 0.0100 |
| | | Median | 0.0299 | 0.0263 | 0.0298 | 0.0194 | 0.0235 |
| $R_{mt}$ | | Maximum | 5.9404 | 3.8786 | 6.9444 | 10.9556 | 3.1536 |
| | | Minimum | −6.9460 | −3.9757 | −6.1243 | −8.9713 | −4.4415 |
| | | Standard deviation | 0.7797 | 0.7356 | 0.6128 | 0.7986 | 0.5393 |
| | Observations | | 3524 | 3471 | 3531 | 3577 | 3563 |

Note: in order to make data readable, all original values are multiplied by 100.

**Table 3.** Chow Test: herding during crisis and non-crisis periods based on different time frequencies.

|  |  |  | Brazil | China | India | Russia | South Africa |
|---|---|---|---|---|---|---|---|
| **Panel A: Sub-Period 1 (02/07/2007–31/03/2009)** | | | | | | | |
| (1) Pre-Global Financial Crisis **(02/07/2007–14/09/2008)** (2) Global Financial Crisis **(15/09/2008–31/03/2009)** | F-statistic (Prob.) | Daily | **52.38 ** (0.00)** | **2.20 *** (0.09)** | **11.45 ** (0.00)** | **122.53 ** (0.00)** | **56.12 ** (0.00)** |
| | | Weekly | **13.26 ** (0.00)** | 0.63 (0.60) | 0.68 (0.57) | **32.79 ** (0.00)** | **8.14 ** (0.00)** |
| | | Monthly | **3.86 ** (0.03)** | **3.86 ** (0.03)** | 0.95 (0.44) | **9.72 ** (0.00)** | 0.86 (0.48) |
| **Panel B: Sub-Period 2 (01/04/2009–31/01/2012)** | | | | | | | |
| (1) Pre-European Debt Crisis **(01/04/2009–31/03/2010)** (2) European Debt Crisis **(01/04/2010–31/01/2012)** | F-statistic (Prob.) | Daily | **10.97 ** (0.00)** | **14.41 ** (0.00)** | **53.23 ** (0.00)** | **57.90 ** (0.00)** | **114.51 ** (0.00)** |
| | | Weekly | **7.61 ** (0.00)** | **4.63 ** (0.00)** | **10.86 ** (0.00)** | **16.53 ** (0.00)** | **26.42 ** (0.00)** |
| | | Monthly | 0.17 (0.91) | 0.69 (0.57) | 1.86 (0.16) | **2.38 *** (0.09)** | **6.85 ** (0.00)** |
| **Panel C: Sub-Period 3 (01/02/2012–30/09/2021)** | | | | | | | |
| (1) Pre COVID-19 Disease Crisis **(01/02/2012–30/12/2019)** (2) COVID-19 Disease Crisis **(31/12/2019–30/09/2021)** | F-statistic (Prob.) | Daily | **40.62 ** (0.00)** | **20.75 ** (0.00)** | **98.44 ** (0.00)** | **10.97 ** (0.00)** | **251.86 ** (0.00)** |
| | | Weekly | **5.19 ** (0.00)** | **4.68 ** (0.00)** | **15.88 ** (0.00)** | **3.21 ** (0.02)** | **55.39 ** (0.00)** |
| | | Monthly | **3.03 ** (0.03)** | **10.11 ** (0.00)** | **7.84 ** (0.00)** | 0.86 (0.47) | **10.91 ** (0.00)** |

Note: This table presents results of $CSAD_t = a_1 + a_2|R_{m,t}| + a_3(R_{m,t})^2 + e_t$. ** and bold numbers mean significant at 5% level. Additionally, the *** and bold numbers represent significant at 10% level.

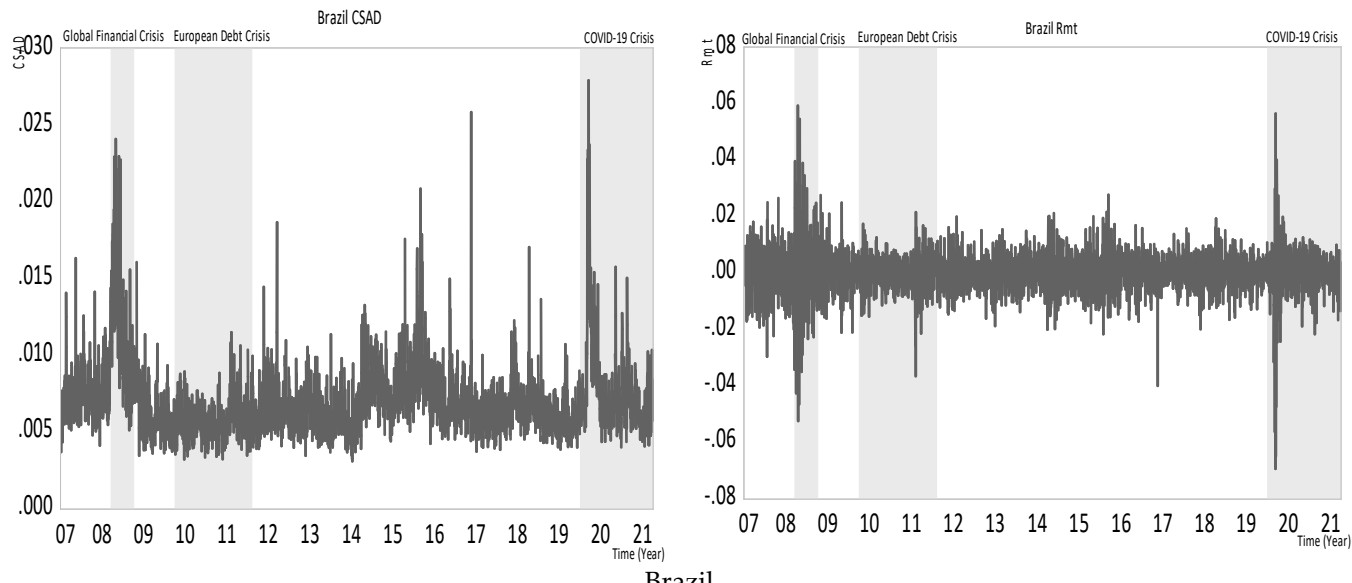

**Figure 1.** *Cont.*

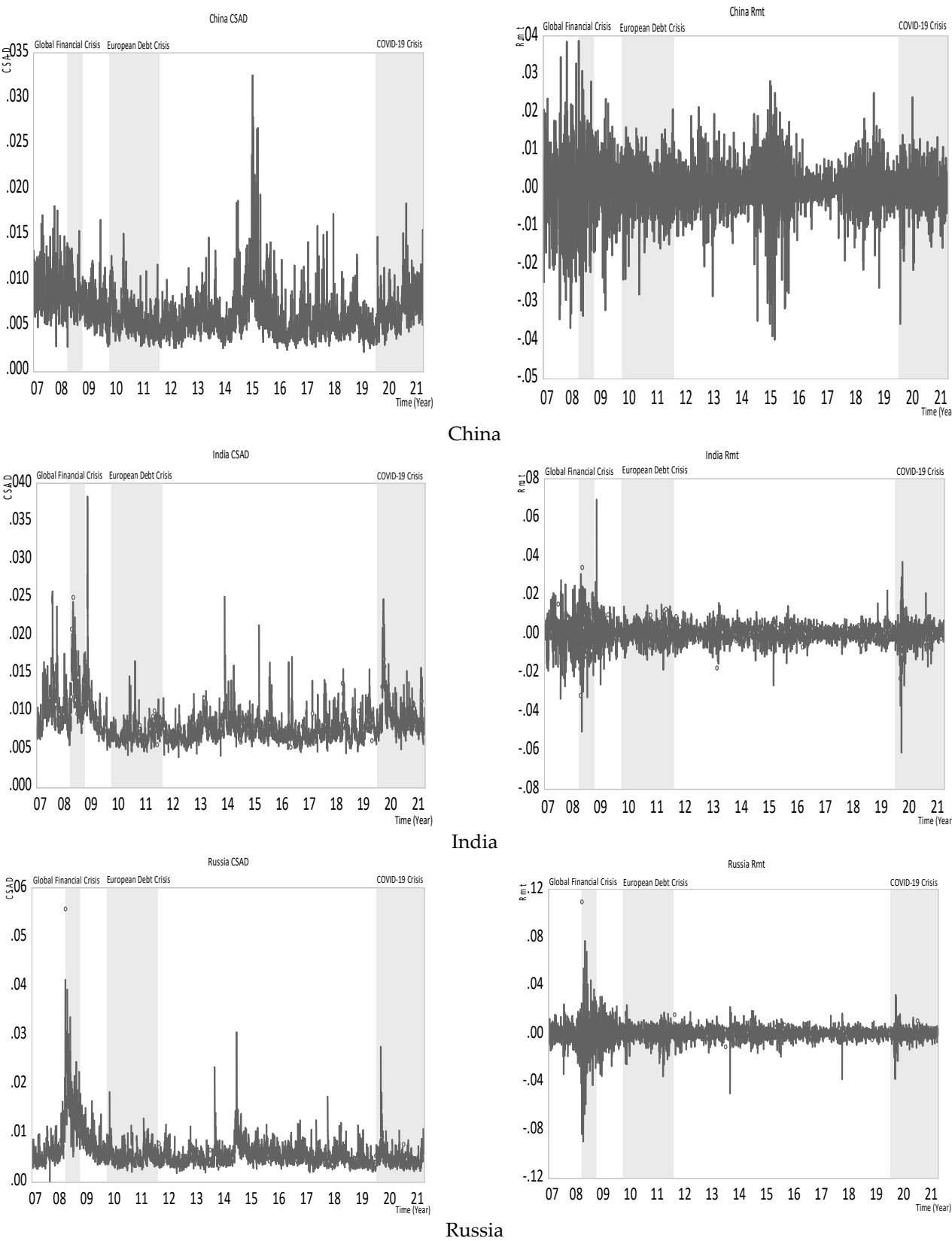

**Figure 1.** *Cont.*

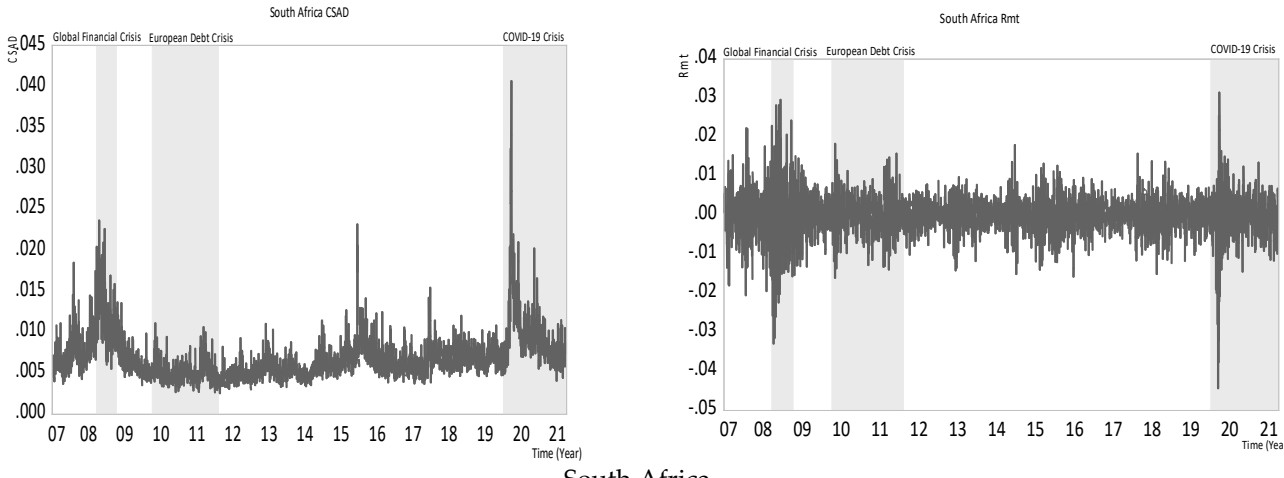

**Figure 1.** CSAD and market return movements based on time-series data. (Note: The shaded areas capture the three different crisis sub-periods. The first shaded area captures results during the Global Financial Crisis from 15 September 2008 to 31 March 2009. The second shaded area captures the European Sovereign Debt Crisis between 1 April 2010 and 31 January 2012. The third shaded area captures the COVID-19 Crisis from 31 December 2019 to 30 April 2020. Furthermore, the X-axis captures "Time". The Y-axis for all graphs captures "CSAD" (**left**) and "Market Return ($R_{mt}$)" (**right**) respectively).

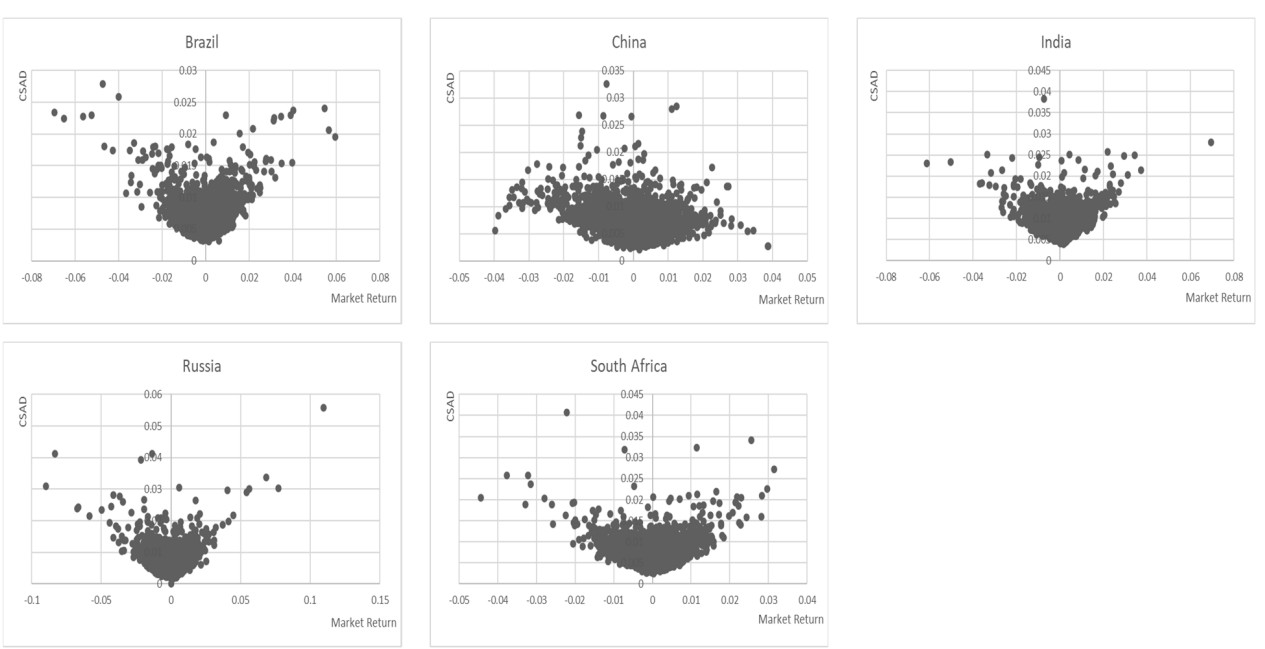

**Figure 2.** Relationship between *CSAD* and market return.

Table 4 presents results on the relationship between herding and different crisis events. Comparing the results in a non-crisis and crisis period, herding is present only in China during the global financial crisis. Surprisingly the global financial crisis, a milestone event, causes herding only in China. The absolute value of the $R_{mt}^2$ coefficient is $|-5.52|$ with a *p*-value of 0.02. The European debt crisis and the COVID-19 Crisis does not lead to a herding effect in any country, based on all countries having a positive coefficient for $R_{mt}^2$ or insignificantly negative coefficient for $R_{mt}^2$. The herding effect in China could be due to the psychology safeguard mechanisms present in investors' mind(s), who wish to avoid making poor decisions. The motivation behind this behaviour can be explained by

"flight to safety", "prospect theory" and "regret aversion behavioural bias". For details, see Kahneman and Tversky (1979), Gazel (2015), Arlen and Tontrup (2016) and Baele et al. (2020). It is encouraging to compare this finding with that found by Chiang and Zheng (2010), who noted that higher levels of herding could be seen in Asian markets during crisis and stress periods. Our hull hypothesis that crises cause herding is not supported. Herding is independent of crises or not. China is the only exception.

**Table 4.** Herding effects using different time frequencies pre- and during crises.

| | | | Brazil | China | India | Russia | South Africa |
|---|---|---|---|---|---|---|---|
| **Panel A: Non-Crisis Period** | | | | | | | |
| Pre-Global Financial Crisis (02/07/2007–14/09/2008) | Daily | $R_{mt}^2$ (Prob.) | 0.79 (0.72) | −2.52 (0.18) | 12.98 (0.00) | −0.44 (0.79) | 9.34 (0.01) |
| | | R-square | 0.21 | 0.02 | 0.29 | 0.42 | 0.41 |
| | Weekly | $R_{mt}^2$ (Prob.) | 1.35 (0.71) | −1.60 (0.52) | 4.65 (0.49) | 9.47 (0.01) | 24.00 (0.00) |
| | | R-square | 0.02 | 0.03 | 0.02 | 0.31 | 0.34 |
| | Monthly | $R_{mt}^2$ (Prob.) | 4.38 (0.51) | 0.33 (0.90) | −7.74 (0.24) | 1.84 (0.64) | 1.37 (0.91) |
| | | R-square | 0.25 | 0.04 | 0.31 | 0.46 | 0.31 |
| Pre-European Debt Crisis (01/04/2009–31/03/2010) | Daily | $R_{mt}^2$ (Prob.) | 8.11 (0.00) | 1.16 (0.63) | 0.82 (0.47) | 2.77 (0.26) | 16.36 (0.01) |
| | | R-square | 0.40 | 0.10 | 0.21 | 0.19 | 0.52 |
| | Weekly | $R_{mt}^2$ (Prob.) | 17.10 (0.00) | −5.09 (0.53) | 9.46 (0.01) | 4.86 (0.19) | 2.15 (0.83) |
| | | R-square | 0.46 | 0.07 | 0.47 | 0.07 | 0.20 |
| | Monthly | $R_{mt}^2$ (Prob.) | 4.66 (0.25) | −1.15 (0.35) | 7.86 (0.00) | −0.71 (0.92) | 4.97 (0.67) |
| | | R-square | 0.57 | 0.81 | 0.81 | 0.46 | 0.13 |
| Pre COVID-19 Disease Crisis (01/02/2012–30/12/2019) | Daily | $R_{mt}^2$ (Prob.) | 11.54 (0.00) | −0.72 (0.53) | 15.30 (0.00) | 2.99 (0.00) | 11.18 (0.00) |
| | | R-square | 0.30 | 0.16 | 0.12 | 0.24 | 0.18 |
| | Weekly | $R_{mt}^2$ (Prob.) | 6.15 (0.00) | 6.28 (0.00) | 0.21 (0.97) | 6.17 (0.01) | 4.03 (0.37) |
| | | R-square | 0.32 | 0.24 | 0.08 | 0.21 | 0.15 |
| | Monthly | $R_{mt}^2$ (Prob.) | 5.17 (0.04) | 4.42 (0.00) | 5.79 (0.35) | 1.41 (0.59) | 14.87 (0.08) |
| | | R-square | 0.33 | 0.45 | 0.02 | 0.15 | 0.03 |
| **Panel B: Crisis Period** | | | | | | | |
| Global Financial Crisis (15/09/2008–31/03/2009) | Daily | $R_{mt}^2$ (Prob.) | 0.43 (0.70) | **−5.52 ** (0.02)** | 3.13 (0.10) | 1.51 (0.04) | 4.84 (0.04) |
| | | R-square | 0.55 | **0.12** | 0.44 | 0.55 | 0.69 |
| | Weekly | $R_{mt}^2$ (Prob.) | 0.17 (0.94) | 3.39 (0.24) | 6.30 (0.02) | −0.46 (0.59) | 5.53 (0.03) |
| | | R-square | 0.45 | 0.08 | 0.45 | 0.59 | 0.70 |
| | Monthly | $R_{mt}^2$ (Prob.) | −2.68 (0.56) | 1.98 (0.13) | 4.69 (0.22) | 6.63 (0.13) | 14.33 (0.06) |
| | | R-square | 0.30 | 0.85 | 0.77 | 0.60 | 0.82 |

**Table 4.** *Cont.*

|  |  |  | Brazil | China | India | Russia | South Africa |
|---|---|---|---|---|---|---|---|
|  |  | **Panel B: Crisis Period** |  |  |  |  |  |
| European Debt Crisis **(01/04/2010– 31/01/2012)** | Daily | $R_{mt}{}^2$ (Prob.) | 1.26 (0.27) | 5.99 (0.02) | 3.87 (0.37) | 2.56 (0.13) | 9.69 (0.00) |
|  |  | R-square | 0.32 | 0.09 | 0.14 | 0.25 | 0.65 |
|  | Weekly | $R_{mt}{}^2$ (Prob.) | −1.28 (0.51) | 17.44 (0.00) | −0.91 (0.88) | 0.07 (0.97) | 2.72 (0.31) |
|  |  | R-square | 0.36 | 0.40 | 0.13 | 0.26 | 0.48 |
|  | Monthly | $R_{mt}{}^2$ (Prob.) | 5.46 (0.24) | 4.76 (0.36) | 6.16 (0.25) | 12.86 (0.04) | 2.44 (0.56) |
|  |  | R-square | 0.33 | 0.25 | 0.15 | 0.26 | 0.36 |
| COVID-19 Disease Crisis **(31/12/2019– 30/09/2021)** | Daily | $R_{mt}{}^2$ (Prob.) | −0.51 (0.39) | 0.52 (0.82) | −0.87 (0.27) | 5.19 (0.00) | −0.13 (0.95) |
|  |  | R-square | 0.54 | 0.11 | 0.48 | 0.51 | 0.39 |
|  | Weekly | $R_{mt}{}^2$ (Prob.) | 1.90 (0.05) | 12.92 (0.03) | 3.73 (0.19) | 3.16 (0.03) | −2.59 (0.46) |
|  |  | R-square | 0.60 | 0.12 | 0.24 | 0.51 | 0.31 |
|  | Monthly | $R_{mt}{}^2$ (Prob.) | 2.15 (0.01) | 2.03 (0.85) | 0.55 (0.78) | 2.96 (0.59) | 31.44 (0.00) |
|  |  | R-square | 0.70 | 0.03 | 0.52 | 0.28 | 0.74 |

Note: This table presents results of $CSAD_t = a_1 + a_2|R_{m,t}| + a_3(R_{m,t})^2 + e_t$. Panel A and B illustrate regression results of non-crisis period and crisis period, respectively. If the coefficient of $R_{mt}{}^2$ is significant and negative, the result can be interpreted as herding, otherwise no herding. ** and bold numbers mean significant at 5% level.

Table 4 also presents regression results of herding when different time-frequencies and crises are considered. With regard to the effect of different time frequencies, what stands out is that only China presents herding (on the basis of daily data), captured by a significantly negative coefficient $R_{mt}{}^2$, however, this phenomenon disappears when other time frequencies are used (as shown by an insignificant or positive coefficient of $R_{mt}{}^2$). This result signifies that the usage of daily data is more likely to capture the short-lived effect of herding. The finding also corroborates the ideas of Banerjee (1992), who suggested that herding tends to be a short-lived effect, resulting in it being relatively difficult to detect using low-frequency data or long horizons, such as using monthly data or yearly data. Our null hypothesis that herding is a short-lived phenomenon is supported.

*4.2. Volatility and Herding*

4.2.1. The Effect of Herding on Volatility

As far as volatility indicators themselves are concerned, the effect of herding on volatility is an open issue (see Blasco et al. (2012) and Alemanni and Ornelas (2006)). Moreover, the results of Table 5 show that there is a positive relationship between volatility measures and trading volume (as shown by the positive coefficients of volume) similar to Blasco et al. (2012). There seems to be no day-of-the-week effect.

**Table 5.** The effects of trading volume and day-of-the-week on volatility measures.

|  |  | Brazil | China | India | Russia | South Africa |
|---|---|---|---|---|---|---|
| $\eta_{\text{GARCH (t)}}$ | $M_t$ | $-1.74 \times 10^{-5}$ | $-2.79 \times 10^{-5}$ | $8.16 \times 10^{-5}$ | $-4.32 \times 10^{-5}$ | 0.0002 |
|  | (Prob.) | (0.89) | (0.83) | (0.50) | (0.82) | (0.05) |
|  | $V_t$ | 0.0028 | 0.0004 | 0.50 | 0.0002 | 0.04 |
|  | (Prob.) | (0.00) | (0.00) | (0.00) | (0.00) | (0.00) |
|  | R-square | 0.0031 | 0.0096 | 0.16 | 0.01 | 0.03 |
| $\eta_{\text{EWMA (t)}}$ | $M_t$ | 0.0010 | $-0.0004$ | 0.0013 | $-0.0005$ | 0.0029 |
|  | (Prob.) | (0.67) | (0.85) | (0.49) | (0.89) | (0.08) |
|  | $V_t$ | 0.05 | 0.0058 | 8.27 | 0.0025 | 0.64 |
|  | (Prob.) | (0.00) | (0.00) | (0.00) | (0.00) | (0.00) |
|  | R-square | 0.0027 | 0.0096 | 0.16 | 0.01 | 0.03 |
| $\eta_{\text{P(t)}}$ | $M_t$ | 0.0001 | 0.0010 | 0.0005 | 0.0004 | 0.0008 |
|  | (Prob.) | (0.69) | (0.00) | (0.08) | (0.37) | (0.00) |
|  | $V_t$ | 0.02 | 0.0022 | 1.10 | 0.0007 | 0.14 |
|  | (Prob.) | (0.00) | (0.00) | (0.00) | (0.00) | (0.00) |
|  | R-square | 0.02 | 0.06 | 0.16 | 0.04 | 0.06 |
| $\eta_{\text{GK (t)}}$ | $M_t$ | $-0.0004$ | 0.0002 | 0.0003 | $-4.70 \times 10^{-5}$ | 0.0005 |
|  | (Prob.) | (0.23) | (0.46) | (0.28) | (0.91) | (0.01) |
|  | $V_t$ | 0.02 | 0.0022 | 1.06 | 0.0007 | 0.12 |
|  | (Prob.) | (0.00) | (0.00) | (0.00) | (0.00) | (0.00) |
|  | R-square | 0.02 | 0.06 | 0.17 | 0.04 | 0.06 |
| $\eta_{\text{MV (t)}}$ | $M_t$ | $6.96 \times 10^{-6}$ | $2.13 \times 10^{-5}$ | $9.74 \times 10^{-5}$ | $-8.65 \times 10^{-6}$ | 0.0002 |
|  | (Prob.) | (0.97) | (0.88) | (0.46) | (0.97) | (0.10) |
|  | $V_t$ | 0.0025 | 0.0004 | 0.53 | 0.0002 | 0.04 |
|  | (Prob.) | (0.02) | (0.00) | (0.00) | (0.00) | (0.00) |
|  | R-square | 0.0017 | 0.01 | 0.15 | 0.01 | 0.02 |
| $\eta_{\text{CW (t)}}$ | $V_t$ | 0.0004 | $6.42 \times 10^{-5}$ | 0.05 | $1.30 \times 10^{-5}$ | 0.01 |
|  | (Prob.) | (0.12) | (0.00) | (0.00) | (0.12) | (0.00) |
|  | R-square | 0.01 | 0.09 | 0.34 | 0.01 | 0.19 |

Note: This table demonstrates how trading volume and day-of-the-week exert influence on volatilities based on the following models $\sigma_{it} = \alpha + \beta M_t + \gamma V_t + \eta_t$ and $\sigma_{CW(t)} = \alpha + \gamma V_t + \eta_t$. All volatilities measures utilised are given here: $R_{mt} = \alpha + \beta R_{m(t-1)} + \varepsilon_t$, $\sigma_{GARCH(t)} = \sqrt{\alpha + \beta \sigma_{Garch\ (t-1)}^2 + \delta \varepsilon_{t-1}^2 + \eta}$, $\sigma_{EWMA(t)} = \sqrt{\lambda\ \sigma_{EWMA(t-1)}^2 + (1-\lambda) R_{mt}^2}$, $\sigma_{P(t)} = \sqrt{\frac{1}{4\sqrt{\ln 2}} \times \frac{1}{n} \sum_{t=1}^n P_t^2}$, $\sigma_{GK(t)} = \sqrt{\frac{1}{n} \sum_{t=1}^n \left[\frac{1}{2} P_t^2 - (2 \ln 2 - 1) Q_t^2\right]}$, $\sigma_{MA\ (t)} = \sqrt{\sum_{t=1}^{20} \left(\frac{1}{20} \times R_{m\ (t-1)}^2\right)}$, $\sigma_{ICW\ (t)} = \sqrt{\frac{\sum_{t=1}^T (R_{it} - \overline{R})^2}{T}}$, $\sigma_{CW\ (t)} = \sum_{i=1}^n \frac{Capitalisation\ i}{total\ capitalisation} \sigma_{ICW\ (t)}$, respectively.

Table 6 presents Chow test results between volatilities and herding. The null hypothesis is rejected so dividing the whole sample into different sub-periods is appropriate. Table 7 shows that the effect of *CSAD* on volatilities is positive (a sign reversal is observed but this is peculiar to a specific period (pre-COVID) and unremarkable) regardless of crisis or not. To explain, low *CSAD* implies herding (movement in a specific direction since opinions converge) which brings about less volatility. Low *CSAD* means that investors tend to behave in a similar way, implying herding. A positive relationship between volatility and *CSAD* indicates that a decrease in *CSAD* (convergence in opinions) results in a decline in volatility (the lower the CSAD, the lower the volatility), and vice versa; greater

herding, less volatility. An increase in *CSAD* means that "individual returns deviate from the market return" (Chang et al. 2000), this means lower herding or diversity in opinions which results in greater volatility. The null hypothesis (greater herding or low *CSAD* causes more volatility in BRICS) is not supported. This outcome is contrary to that of Blasco et al. (2012) for the Spanish market which showed that greater herding triggers volatility. In their research, Blasco et al. (2012) explored the relationship between herding intensity and different volatility measures. According to their research, increased herding triggers an increase in volatility measures, except implied volatility. In comparison, Alemanni and Ornelas (2006) observed that herding did not explain volatility movements in their research on nine emerging markets, including Brazil, India and South Africa.

**Table 6.** Chow Test: The Effect of *CSAD* on Volatilities based on Crisis and Non-crisis period(s).

| | | | Brazil | China | India | Russia | South Africa |
|---|---|---|---|---|---|---|---|
| **Panel A: Sub-Period 1 (02/07/2007–31/03/2009)** | | | | | | | |
| (1) Pre-Global Financial Crisis (02/07/2007–14/09/2008) (2) Global Financial Crisis (15/09/2008–31/03/2009) | F-statistic (Prob.) | $\eta$GARCH | **124.07 \*\*** **(0.00)** | **13.60 \*\*** **(0.00)** | **75.83 \*\*** **(0.00)** | **91.25 \*\*** **(0.00)** | **106.13 \*\*** **(0.00)** |
| | | $\eta$Ewma | **212.79 \*\*** **(0.00)** | **12.60 \*\*** **(0.00)** | **106.62 \*\*** **(0.00)** | **164.59 \*\*** **(0.00)** | **163.11 \*\*** **(0.00)** |
| | | $\eta$P | **40.45 \*\*** **(0.00)** | 1.40 (0.25) | **22.82 \*\*** **(0.00)** | **12.33 \*\*** **(0.00)** | **13.41 \*\*** **(0.00)** |
| | | $\eta$GK | **19.45 \*\*** **(0.00)** | 2.03 (0.13) | **11.55 \*\*** **(0.00)** | 1.15 (0.32) | **16.93 \*\*** **(0.00)** |
| | | $\eta$MA | **144.39 \*\*** **(0.00)** | **10.27 \*\*** **(0.00)** | **66.00 \*\*** **(0.00)** | **89.78 \*\*** **(0.00)** | **99.68 \*\*** **(0.00)** |
| | | $\eta$CW | **4.96 \*\*** **(0.02)** | 0.15 (0.86) | **9.51 \*\*** **(0.00)** | 1.28 (0.30) | **11.93 \*\*** **(0.00)** |
| **Panel B: Sub-Period 2 (01/04/2009–31/01/2012)** | | | | | | | |
| (1) Pre-European Debt Crisis (01/04/2009–31/03/2010) (2) European Debt Crisis (01/04/2010–31/01/2012) | F-statistic (Prob.) | $\eta$GARCH | **3.68 \*\*** **(0.03)** | **30.67 \*\*** **(0.00)** | **97.47 \*\*** **(0.00)** | **28.04 \*\*** **(0.00)** | **13.85 \*\*** **(0.00)** |
| | | $\eta$Ewma | **17.12 \*\*** **(0.00)** | **34.84 \*\*** **(0.00)** | **90.55 \*\*** **(0.00)** | **55.55 \*\*** **(0.00)** | **25.60 \*\*** **(0.00)** |
| | | $\eta$P | **27.61 \*\*** **(0.00)** | **106.09 \*\*** **(0.00)** | 1.69 (0.18) | 0.15 (0.86) | **27.61 \*\*** **(0.00)** |
| | | $\eta$GK | 1.77 (0.17) | **13.22 \*\*** **(0.00)** | 2.08 (0.13) | 1.39 (0.25) | **8.11 \*\*** **(0.00)** |
| | | $\eta$MA | **2.74 \*\*\*** **(0.07)** | **16.50 \*\*** **(0.00)** | **72.31 \*\*** **(0.00)** | **29.95 \*\*** **(0.00)** | **20.81 \*\*** **(0.00)** |
| | | $\eta$CW | 0.86 (0.43) | **2.57 \*\*\*** **(0.09)** | **7.50 \*\*** **(0.00)** | 0.30 (0.82) | **4.01 \*\*** **(0.03)** |
| **Panel C: Sub-Period 3 (01/02/2012–30/09/2021)** | | | | | | | |
| (1) Pre COVID-19 Crisis (01/02/2012–30/12/2019) (2) COVID-19 Crisis (31/12/2019–30/09/2021) | F-statistic (Prob.) | $\eta$GARCH | **871.35 \*\*** **(0.00)** | **51.17 \*\*** **(0.00)** | **350.56 \*\*** **(0.00)** | **142.38 \*\*** **(0.00)** | **413.30 \*\*** **(0.00)** |
| | | $\eta$Ewma | **906.32 \*\*** **(0.00)** | **903.22 \*\*** **(0.00)** | **268.73 \*\*** **(0.00)** | **121.54 \*\*** **(0.00)** | **377.22 \*\*** **(0.00)** |
| | | $\eta$P | **18.90 \*\*** **(0.00)** | **28.07 \*\*** **(0.00)** | **236.99 \*\*** **(0.00)** | **40.88 \*\*** **(0.00)** | **95.15 \*\*** **(0.00)** |
| | | $\eta$GK | **19.08 \*\*** **(0.00)** | **39.61 \*\*** **(0.00)** | **222.95 \*\*** **(0.00)** | **52.88 \*\*** **(0.00)** | **97.69 \*\*** **(0.00)** |
| | | $\eta$MA | **564.70 \*\*** **(0.00)** | **41.57 \*\*** **(0.00)** | **263.44 \*\*** **(0.00)** | **98.24 \*\*** **(0.00)** | **341.37 \*\*** **(0.00)** |
| | | $\eta$CW | **61.64 \*\*** **(0.00)** | **3.42 \*\*** **(0.04)** | **7.12 \*\*** **(0.00)** | **19.94 \*\*** **(0.00)** | **14.28 \*\*** **(0.00)** |

Note: The results from this table are based on $\eta_t = \alpha + \beta CSAD_t + \varepsilon_t$. \*\* and bold numbers mean significant at 5% level and \*\*\* and bold numbers stands for significant at 10% level.

**Table 7.** The Effect of *CSAD* on Volatilities.

| | | | Brazil | China | India | Russia | South Africa |
|---|---|---|---|---|---|---|---|
| | | **Panel A: Non-Crisis Period** | | | | | |
| Pre-Global Financial Crisis (02/07/2007–14/09/2008) | $\eta_{GARCH}$ | CSAD (Prob.) | **0.08 ** (0.04)** | −0.04 (0.52) | **0.08 *** (0.06)** | **0.44 ** (0.00)** | **0.26 ** (0.00)** |
| | $\eta_{Ewma}$ | CSAD (Prob.) | **1.56 ** (0.01)** | −0.57 ** (0.51) | 0.04 (0.95) | **0.14 ** (0.04)** | **3.79 ** (0.00)** |
| | $\eta_P$ | CSAD (Prob.) | **1.02 ** (0.00)** | **0.85 ** (0.00)** | **0.92 ** (0.00)** | **2.16 ** (0.00)** | **1.12 ** (0.00)** |
| | $\eta_{GK}$ | CSAD (Prob.) | **0.84 ** (0.00)** | **0.89 ** (0.00)** | **0.82 ** (0.00)** | **1.86 ** (0.00)** | **0.56 ** (0.00)** |
| | $\eta_{MA}$ | CSAD (Prob.) | −0.03 (0.47) | −0.08 (0.23) | 0.05 (0.27) | **0.39 ** (0.00)** | **0.26 ** (0.00)** |
| | $\eta_{CW}$ | CSAD (Prob.) | −0.03 (0.75) | 0.12 (0.12) | **0.10 ** (0.00)** | **0.14 ** (0.04)** | **0.10 ** (0.00)** |
| Pre-European Debt Crisis (01/04/2009–31/03/2010) | $\eta_{GARCH}$ | CSAD (Prob.) | **0.09 ** (0.01)** | 0.04 (0.51) | **0.59 ** (0.00)** | **0.47 ** (0.00)** | **0.34** (0.00) |
| | $\eta_{Ewma}$ | CSAD (Prob.) | **3.28 ** (0.00)** | 0.49 (0.58) | **8.84 ** (0.00)** | **8.72 ** (0.00)** | **6.85 ** (0.00)** |
| | $\eta_P$ | CSAD (Prob.) | **1.03 ** (0.00)** | **0.51** (0.00) | **0.62 ** (0.00)** | **1.10 ** (0.00)** | **1.12 ** (0.00)** |
| | $\eta_{GK}$ | CSAD (Prob.) | **1.39 ** (0.00)** | **1.06 ** (0.00)** | **0.50 ** (0.00)** | **0.75 ** (0.00)** | **0.79 ** (0.00)** |
| | $\eta_{MA}$ | CSAD (Prob.) | **0.08 ** (0.05)** | 0.03 (0.70) | **0.58 ** (0.00)** | **0.54 ** (0.00)** | **0.44 ** (0.00)** |
| | $\eta_{CW}$ | CSAD (Prob.) | −0.0025 (0.97) | **0.08 *** (0.08)** | **0.15 ** (0.00)** | 0.07 (0.25) | **0.24 ** (0.00)** |
| Pre COVID-19 Crisis (01/02/2012–30/12/2019) | $\eta_{GARCH}$ | CSAD (Prob.) | **−0.03 ** (0.00)** | **0.18 ** (0.00)** | **−0.03 *** (0.09)** | **−0.03 ** (0.01)** | −0.01 (0.22) |
| | $\eta_{Ewma}$ | CSAD (Prob.) | **−0.93 ** (0.00)** | **2.79 ** (0.00)** | **−0.54 ** (0.05)** | **−0.78 ** (0.00)** | **−0.62 ** (0.00)** |
| | $\eta_P$ | CSAD (Prob.) | **0.41 ** (0.00)** | **0.51 ** (0.00)** | **0.13 ** (0.00)** | **0.33 ** (0.00)** | **0.16 ** (0.00)** |
| | $\eta_{GK}$ | CSAD (Prob.) | **0.30 ** (0.00)** | **0.55 ** (0.00)** | 0.06 (0.14) | **0.39 ** (0.00)** | **0.11 ** (0.00)** |
| | $\eta_{MA}$ | CSAD (Prob.) | **−0.05 ** (0.00)** | **0.19 ** (0.00)** | **−0.04 ** (0.01)** | **−0.04 ** (0.00)** | **−0.02 ** (0.04)** |
| | $\eta_{CW}$ | CSAD (Prob.) | **0.04 ** (0.04)** | 0.02 (0.24) | 0.0015 (0.93) | −0.02 (0.25) | **0.04 ** (0.04)** |
| | | **Panel B: Crisis Period** | | | | | |
| Global Financial Crisis (15/09/2008–31/03/2009) | $\eta_{GARCH}$ | CSAD (Prob.) | **0.91 ** (0.00)** | **0.28 ** (0.00)** | **0.58 ** (0.00)** | **0.57 ** (0.00)** | **0.30 ** (0.00)** |
| | $\eta_{Ewma}$ | CSAD (Prob.) | **15.92 ** (0.00)** | **4.17 ** (0.00)** | **7.13 ** (0.00)** | **7.23 ** (0.00)** | **3.80 ** (0.00)** |
| | $\eta_P$ | CSAD (Prob.) | **3.47 ** (0.00)** | **1.28 ** (0.00)** | **1.95 ** (0.00)** | **3.06 ** (0.00)** | **1.89 ** (0.00)** |
| | $\eta_{GK}$ | CSAD (Prob.) | **2.42 ** (0.00)** | **1.52 ** (0.00)** | **1.65 ** (0.00)** | **2.22 ** (0.00)** | **0.79 ** (0.00)** |
| | $\eta_{MA}$ | CSAD (Prob.) | **1.10 ** (0.00)** | **0.29 ** (0.00)** | **0.57 ** (0.00)** | **0.61** (0.00) | **0.33 ** (0.00)** |
| | $\eta_{CW}$ | CSAD (Prob.) | **0.44 ** (0.05)** | 0.20 (0.28) | **0.32 ** (0.00)** | 0.32 (0.12) | 0.09 (0.63) |

**Table 7.** *Cont.*

|  |  |  | Brazil | China | India | Russia | South Africa |
|---|---|---|---|---|---|---|---|
|  |  | **Panel B: Crisis Period** |  |  |  |  |  |
| European Debt Crisis (01/04/2010–31/01/2012) | $\eta$GARCH | CSAD (Prob.) | **0.11 **** **(0.00)** | 0.02 (0.25) | −0.02 (0.44) | **0.19 **** **(0.00)** | **0.17 **** **(0.00)** |
|  | $\eta$Ewma | CSAD (Prob.) | **1.03 ***** **(0.07)** | 0.24 (0.39) | −0.05 (0.88) | **2.50 **** **(0.00)** | **2.32 **** **(0.00)** |
|  | $\eta$P | CSAD (Prob.) | **0.63 **** **(0.00)** | **0.45 **** **(0.00)** | **0.40 **** **(0.00)** | **1.19 **** **(0.00)** | **1.08 **** **(0.00)** |
|  | $\eta$GK | CSAD (Prob.) | **0.34 **** **(0.01)** | **0.37 **** **(0.00)** | **0.28 **** **(0.00)** | **1.03 **** **(0.00)** | **0.45 **** **(0.00)** |
|  | $\eta$MA | CSAD (Prob.) | 0.06 (0.17) | **0.04 **** **(0.05)** | −0.03 (0.28) | **0.18 **** **(0.00)** | **0.17 **** **(0.00)** |
|  | $\eta$CW | CSAD (Prob.) | −0.08 (0.14) | −0.02 (0.47) | **−0.01 **** (0.61) | **0.12 **** **(0.04)** | **0.01 **** (0.86) |
| COVID-19 Crisis (31/12/2019–30/09/2021) | $\eta$GARCH | CSAD (Prob.) | **1.06 **** **(0.00)** | −0.01 (0.70) | **0.90 **** **(0.00)** | **0.30 **** **(0.00)** | **0.46 **** **(0.00)** |
|  | $\eta$Ewma | CSAD (Prob.) | **18.32 **** **(0.00)** | −0.14 (0.65) | **12.62 **** **(0.00)** | **3.49 **** **(0.00)** | **7.27 **** **(0.00)** |
|  | $\eta$P | CSAD (Prob.) | **2.45 **** **(0.00)** | −0.07 (0.24) | **1.84 **** **(0.00)** | **0.75 **** **(0.00)** | **0.79 **** **(0.00)** |
|  | $\eta$GK | CSAD (Prob.) | **2.09 **** **(0.00)** | **−0.16** **(0.00)** | **1.70 **** **(0.00)** | **0.96 **** **(0.00)** | **0.67 **** **(0.00)** |
|  | $\eta$MA | CSAD (Prob.) | **1.15 **** **(0.00)** | **−0.05 **** **(0.02)** | **0.86 **** **(0.00)** | **0.29 **** **(0.00)** | **0.49 **** **(0.00)** |
|  | $\eta$CW | CSAD (Prob.) | **0.48 **** **(0.00)** | **−0.04 **** (0.15) | **0.16 **** **(0.00)** | **0.25 **** **(0.00)** | **0.12 **** **(0.00)** |

Note: The results from this table are based on $\eta_t = \alpha + \beta CSAD_t + \varepsilon_t$. ** and bold numbers mean significant at 5% level as well as *** and bold numbers stands for significant at 10% level.

### 4.2.2. The Effect of Volatility on Herding

Regression results looking at the influence of volatility on the degree of herding are reported in Table 8. Firstly, from a market perspective, it is worth noting that herding is not present in Russia and South Africa regardless of any volatility measure and any volatility level. Conversely, herding is present in Brazil, China and India for the high volatility period (Group 4—shown in Panel D of Table 8). This is captured by the significant negative coefficient of $R_{mt}^2$ in group 4. This finding echoes the results of Zheng et al. (2015) and Lakshman et al. (2013), who observed that herding is more pronounced in the riskier markets of China and India. In other words, a higher volatility can amplify herding, resulting in relatively more irrational investors. This relationship may be partly explained by the fact that volatility can increase investors' anxiety sentiments and hamper their analytical ability and objectivity, resulting in a loss of confidence in their judgement and an increasing tendency to follow the market consensus (Lao and Singh 2011). Finally, regarding the degree of influence that volatility has on herding, China can be regarded as highly susceptible to volatility due to their significant negative coefficient of $R_{mt}^2$ for the majority of the different volatility methods used, excluding $\eta_{CW(t)}$. Specifically, the effect of volatility on herding was stronger in China than that of Brazil and India, based on the greater absolute value of the coefficients in China (such as: |−9.37| of $\eta_{EWMA(t)}$ and |−9.37| of $\eta_{Garch}$ compared to |−1.12| and |−1.11| in India, respectively).

**Table 8.** The effect of different volatility measures and volatility intensity on herding.

| | | Brazil | China | India | Russia | South Africa |
|---|---|---|---|---|---|---|
| **Panel A: Group 1: Volatilities Level $\leq$ 25% of Volatilities Distributions** | | | | | | |
| | | Brazil | China | India | Russia | South Africa |
| $\eta$**GARCH** | $R_{mt}^2$ (Prob.) | 4.28 (0.03) | 3.46 (0.03) | 21.09 (0.00) | 5.21 (0.00) | 10.33 (0.01) |
| | R-square | 0.22 | 0.14 | 0.17 | 0.29 | 0.17 |
| $\eta$**Ewma** | $R_{mt}^2$ (Prob.) | 7.25 (0.01) | 3.81 (0.01) | 10.68 (0.00) | 6.09 (0.00) | 6.04 (0.14) |
| | R-square | 0.15 | 0.15 | 0.10 | 0.28 | 0.16 |
| $\eta$**P** | $R_{mt}^2$ (Prob.) | 19.70 (0.00) | −0.73 (0.65) | 23.16 (0.00) | 26.93 (0.00) | 72.85 (0.00) |
| | R-square | 0.03 | 0.01 | 0.08 | 0.09 | 0.09 |
| $\eta$**GK** | $R_{mt}^2$ (Prob.) | 6.37 (0.03) | −3.46 (0.06) | 6.74 (0.03) | 12.66 (0.00) | 12.64 (0.05) |
| | R-square | 0.14 | 0.05 | 0.11 | 0.16 | 0.15 |
| $\eta$**MA** | $R_{mt}^2$ (Prob.) | 8.05 (0.00) | 3.37 (0.03) | 12.20 (0.00) | 6.10 (0.00) | 7.00 (0.08) |
| | R-square | 0.25 | 0.14 | 0.16 | 0.28 | 0.19 |
| $\eta$**CW** | $R_{mt}^2$ (Prob.) | 1.87 (0.56) | 0.78 (0.84) | −3.57 (0.60) | 3.53 (0.00) | 11.30 (0.12) |
| | R-square | 0.25 | 0.03 | 0.04 | 0.48 | 0.13 |
| **Panel B: Group 2: 25% of Volatilities Distributions < Volatilities Level $\leq$ 50% of Volatilities Distributions** | | | | | | |
| | | Brazil | China | India | Russia | South Africa |
| $\eta$**GARCH** | $R_{mt}^2$ (Prob.) | 9.59 (0.00) | 0.73 (0.70) | 8.34 (0.01) | 0.77 (0.78) | 13.01 (0.01) |
| | R-square | 0.25 | 0.10 | 0.08 | 0.13 | 0.18 |
| $\eta$**Ewma** | $R_{mt}^2$ (Prob.) | 12.80 (0.00) | −0.64 (0.80) | 19.82 (0.00) | 1.33 (0.60) | 12.80 (0.00) |
| | R-square | 0.16 | 0.10 | 0.15 | 0.17 | 0.16 |
| $\eta$**P** | $R_{mt}^2$ (Prob.) | 17.10 (0.00) | 10.93 (0.00) | 8.67 (0.01) | 19.40 (0.00) | 46.74 (0.00) |
| | R-square | 0.07 | 0.13 | 0.09 | 0.09 | 0.13 |
| $\eta$**GK** | $R_{mt}^2$ (Prob.) | 5.75 (0.04) | 0.98 (0.61) | 11.12 (0.00) | 1.82 (0.23) | 27.34 (0.00) |
| | R-square | 0.13 | 0.09 | 0.18 | 0.21 | 0.24 |
| $\eta$**MA** | $R_{mt}^2$ (Prob.) | 0.83 (0.49) | 0.71 (0.74) | 8.47 (0.01) | 4.75 (0.11) | 19.80 (0.00) |
| | R-square | 0.18 | 0.09 | 0.10 | 0.16 | 0.21 |
| $\eta$**CW** | $R_{mt}^2$ (Prob.) | 2.65 (0.43) | 2.05 (0.42) | 7.15 (0.30) | −3.10 (0.42) | 11.59 (0.40) |
| | R-square | 0.10 | 0.15 | 0.03 | 0.10 | 0.03 |

**Table 8.** *Cont.*

| Panel C: Group 3: 50% of Volatilities Distributions < Volatilities Level ≤ 85% of Volatilities Distributions | | | | | | |
|---|---|---|---|---|---|---|
| | | **Brazil** | **China** | **India** | **Russia** | **South Africa** |
| $\eta_{GARCH}$ | $R_{mt}^2$ (Prob.) | 7.65 (0.00) | 0.02 (0.99) | 4.65 (0.02) | 7.59 (0.01) | 8.36 (0.01) |
| | R-square | 0.29 | 0.12 | 0.18 | 0.19 | 0.22 |
| $\eta_{Ewma}$ | $R_{mt}^2$ (Prob.) | 4.52 (0.01) | 0.55 (0.74) | 10.12 (0.00) | 6.27 (0.05) | 10.36 (0.00) |
| | R-square | 0.20 | 0.55 | 0.21 | 0.21 | (0.25) |
| $\eta_P$ | $R_{mt}^2$ (Prob.) | 7.60 (0.04) | −0.24 (0.95) | 9.86 (0.00) | 9.39 (0.06) | 0.79 (0.94) |
| | R-square | 0.19 | 0.11 | 0.23 | 0.18 | 0.14 |
| $\eta_{GK}$ | $R_{mt}^2$ (Prob.) | 6.07 (0.00) | 0.61 (0.84) | 5.85 (0.04) | 6.59 (0.02) | 14.12 (0.00) |
| | R-square | 0.27 | 0.11 | 0.16 | 0.25 | 0.22 |
| $\eta_{MA}$ | $R_{mt}^2$ (Prob.) | 9.91 (0.00) | −1.04 (0.50) | 6.62 (0.00) | 0.62 (0.80) | 7.39 (0.01) |
| | R-square | 0.34 | 0.13 | 0.24 | 0.19 | 0.24 |
| $\eta_{CW}$ | $R_{mt}^2$ (Prob.) | 4.84 (0.31) | 1.03 (0.82) | 14.93 (0.01) | 0.62 (0.80) | −19.10 (0.12) |
| | R-square | 0.18 | 0.16 | 0.31 | 0.19 | 0.06 |
| Panel D: Group 4: 75% of Volatilities Distributions < Volatilities Level ≤ 100% of Volatilities Distributions | | | | | | |
| | | **Brazil** | **China** | **India** | **Russia** | **South Africa** |
| $\eta_{GARCH}$ | $R_{mt}^2$ (Prob.) | **−0.85 \*\*\*** **(0.07)** | **−9.37 \*\*** **(0.00)** | **−1.11 \*\*\*** **(0.09)** | −0.15 (0.65) | 1.09 (0.46) |
| | R-square | **0.57** | **0.19** | **0.44** | 0.56 | 0.50 |
| $\eta_{Ewma}$ | $R_{mt}^2$ (Prob.) | **−1.25 \*\*** **(0.01)** | **−9.37 \*\*** **(0.00)** | **−1.12 \*\*\*** **(0.08)** | −0.20 (0.56) | 0.27 (0.86) |
| | R-square | **0.60** | **0.19** | **0.44** | 0.57 | 0.50 |
| $\eta_P$ | $R_{mt}^2$ (Prob.) | −0.24 (0.56) | **−6.82 \*\*** **(0.00)** | 0.27 (0.67) | 0.12 (0.72) | 5.29 (0.00) |
| | R-square | 0.60 | **0.18** | 0.41 | 0.56 | 0.48 |
| $\eta_{GK}$ | $R_{mt}^2$ (Prob.) | **−0.89 \*\*\*** **(0.06)** | **−8.55 \*\*** **(0.00)** | −0.51 (0.43) | −0.19 (0.59) | 2.40 (0.11) |
| | R-square | **0.58** | **0.20** | 0.44 | 0.55 | 0.49 |
| $\eta_{MA}$ | $R_{mt}^2$ (Prob.) | 1.93 (0.20) | **−9.51 \*\*** **(0.00)** | **−1.05** **(0.10)** | 1.93 (0.20) | 1.93 (0.20) |
| | R-square | 0.50 | **0.20** | **0.44** | 0.50 | 0.50 |
| $\eta_{CW}$ | $R_{mt}^2$ (Prob.) | −0.77 (0.44) | 0.48 (0.74) | 1.86 (0.16) | 3.25 (0.06) | 16.02 (0.03) |
| | R-square | 0.49 | 0.36 | 0.59 | 0.51 | 0.46 |

Note: A quartile regression model has been used to capture the relationship between volatility intensity (high vs low) and herding. All data is divided into four different groups/periods based on volatility magnitude. Group/period 1 is the lowest magnitude group/period and Group/period 4 is the highest magnitude group/period (details shown in the table above) \*\* and bold numbers mean significant at 5% level and \*\*\* and bold numbers mean significant at 10% level.

These findings indicate that our null hypothesis (higher volatility causes more herding or reduces CSAD) is supported. Secondly, from of a volatility measure perspective, what is surprising is that the capitalisation-weighted average volatility (CW) showed no significant effects on investors' behaviour for all the sample markets during both high volatility and low volatility periods. In other words, the effect of volatility on herding appears to be dependent on the different volatility measurement methods. This means that the null hypothesis (volatility measure homogeneity and equal impact on herding) is not supported.

### 4.3. Sentiment Index and Herding

Table 9 presents details on the variables used in principal component analysis. From Table 10, it can be clearly seen that the values obtained from the Bartlett test of sphericity showed that all values were significant. Kaiser (1970) put forward "a Measure of Sampling Adequacy" which can be utilised to test the suitability of sample data for factor analysis by comparing the correlation coefficient and partial correlation coefficient of variables. According to Kaiser (1970, 1974) and Hair et al. (2006), research suggests that KMO values range from 0 to 1 and results greater than 0.5 or significant Bartlett Test of Sphericity are acceptable and suitable for factor analysis. Therefore, all variables are appropriate for PCA.

**Table 9.** Indicators used to measure the Sentiment Index.

|  | Indicator Name | Formula/Symbol |
|---|---|---|
| Market transaction indicator | Turnover | TURN (t) = (turn (t))/(TURNMV5) |
| Market valuation indicator | Price-earnings ratio | PE (t) = Price to earnings ratio |
| Macroeconomic indicators | Consumer price index | GCPI (t) = Log (CPI $_{(t)}$/CPI $_{(t-1)}$) |
|  | Industrial production | GIP (t) = Log (IP $_{(t)}$/IP $_{(t-1)}$) |
|  | Money supply | GM2 (t) = Log (M2 $_{(t)}$/M2 $_{(t-1)}$)/ |
|  | Exchange rate | GER (t) = Log (ER $_{(t)}$/ER $_{(t-1)}$) |

Note: (1) TURN(t) = turnover ratio; (2) TURNMV5 = the average turnover in the previous 5 months; (3) PE(t) = market price–earnings ratio qt month t; (4) GCPI (t) = growth rate of CPI; (5) GIP (t) = growth rate of industrial production; (6) GM2 (t) = change in monthly supply of M2; (7) GER (t) = change in exchange rate of domestic currency relative to the US dollar. With regard to market transaction indicator, both Baker et al. (2012) and Baker and Wurgler (2006) included market turnover in PCA to capture sentiment. With regard to the market valuation, price–earnings ratio (P/E) is seen as an indicator that reflects not only stock price, but also represent companies' earnings ability, which can be utilised to predict corporations' future profitability. According to Khan and Ahmad (2018), weighted P/E ratio can be included in PCA to calculate the sentiment index. Bouteska (2020) also used P/E ratio to construct the sentiment index. Finally, macroeconomic variables are also taken into account because these indicators not only influence the economy, but can also result in fluctuations in investors' emotions. In terms of industrial production, Baker and Wurgler (2007) included it in their research as one of their control variables to build the sentiment index and confirmed the effects of macroeconomic variables on investors' behaviour, especially noisy traders. In addition, according to Chen et al. (2014), money supply and exchange rate are positively related to fluctuations of the sentiment index.

**Table 10.** KMO and Bartlett Result.

|  | Brazil | China | India | Russia | South Africa |
|---|---|---|---|---|---|
| KMO Measure of Sampling Adequacy | 0.5540 | 0.4980 | 0.5270 | 0.5430 | 0.4860 |
| Bartlett Test of Sphericity (Sig.) | 0.0010 | 0.0040 | 0.0300 | 0.0000 | 0.0500 |

In order to construct a sentiment index (*SIX*), five factors were utilised in the PCA. The results are shown in Table 11, which reports the values of the five principal components for every variable and the explanatory power of every principal component. From Table 11, the final sentiment index is captured by the following models for all five sample markets, based on the weighted average method for the first five principal components:

$$SIX_{Brazil} = 0.0320\ TURN\ (t) + 0.1881\ PE\ (t) - 0.0394\ GCPI\ (t) - 0.1165\ GIP\ (t) + 0.3845\ GM2\ (t) + 0.1121\ GER\ (t) \quad (19)$$

$$SIX_{China} = 0.1363\ TURN\ (t) + 0.2266\ PE\ (t) + 0.1102\ GCPI\ (t) + 0.1791\ GIP\ (t) + 0.2909\ GM2\ (t) + 0.0986\ GER\ (t) \quad (20)$$

$$SIX_{India} = 0.3433\ TURN\ (t) + 0.0945\ PE\ (t) + 0.0347\ GCPI\ (t) + 0.1082\ GIP\ (t) + 0.1162\ GM2\ (t) - 0.2316\ GER\ (t) \quad (21)$$

$$SIX_{Russia} = 0.1159\ TURN\ (t) + 0.1359\ PE\ (t) + 0.1147\ GCPI\ (t) + 0.1696\ GIP\ (t) + 0.3578\ GM2\ (t) + 0.1148\ GER\ (t) \quad (22)$$

$$SIX_{South\ Africa} = 0.1564\ TURN\ (t) + 0.1735\ PE\ (t) + 0.0563\ GCPI\ (t) - 0.0482\ GIP\ (t) + 0.3450\ GM2\ (t) + 0.1632\ GER\ (t) \quad (23)$$

**Table 11.** Principal Component (PC) Results: The Effects of various Economic Indicators on Sentiment Index.

| | | PC 1 | PC 2 | PC 3 | PC 4 | PC 5 | PC 6 |
|---|---|---|---|---|---|---|---|
| **Panel A: Brazil** | | | | | | | |
| | TURN(t) | −0.3539 | 0.5736 | −0.1727 | −0.2717 | 0.5539 | 0.3678 |
| | PE(t) | 0.0011 | 0.1688 | 0.9605 | −0.2173 | 0.0341 | −0.0230 |
| Eigenvector | GCPI(t) | 0.1791 | −0.7056 | 0.0730 | −0.1623 | 0.5939 | 0.2926 |
| | GIP(t) | −0.5449 | −0.1937 | 0.1689 | 0.4949 | −0.2370 | 0.5796 |
| | GM2(t) | 0.4355 | 0.2789 | 0.1061 | 0.7512 | 0.3961 | −0.0078 |
| | GER(t) | 0.5967 | 0.1711 | −0.0492 | −0.2083 | −0.3553 | 0.6653 |
| Eigenvalues | | 1.4891 | 1.2080 | 1.0004 | 0.8766 | 0.7498 | 0.6762 |
| Proportion | | 0.2482 | 0.2013 | 0.1667 | 0.1461 | 0.1250 | 0.1127 |
| Cumulative Proportion | | 0.2482 | 0.4495 | 0.6162 | 0.7623 | 0.8873 | 1.0000 |
| **Panel B: China** | | | | | | | |
| | TURN(t) | 0.1019 | 0.5722 | 0.1350 | −0.6527 | 0.4558 | 0.1013 |
| | PE(t) | 0.6720 | −0.0109 | 0.0887 | 0.2215 | −0.0014 | 0.7010 |
| Eigenvector | GCPI(t) | 0.4809 | −0.4878 | 0.1484 | −0.0267 | 0.5289 | −0.4779 |
| | GIP(t) | 0.0852 | 0.4759 | −0.5339 | 0.5543 | 0.3755 | −0.1811 |
| | GM2(t) | 0.2477 | 0.4516 | 0.6279 | 0.2727 | −0.3296 | −0.3968 |
| | GER(t) | −0.4880 | −0.0638 | 0.5221 | 0.3777 | 0.5126 | 0.2825 |
| Eigenvalues | | 1.3811 | 1.2405 | 1.0332 | 0.9304 | 0.7588 | 0.6561 |
| Proportion | | 0.2302 | 0.2068 | 0.1722 | 0.1551 | 0.1265 | 0.1093 |
| Cumulative Proportion | | 0.2302 | 0.4369 | 0.6091 | 0.7642 | 0.8907 | 1.0000 |
| **Panel C: India** | | | | | | | |
| | TURN(t) | 0.1996 | 0.4706 | 0.6300 | 0.5560 | −0.1112 | 0.1425 |
| | PE(t) | 0.4602 | 0.0780 | 0.3014 | −0.6714 | 0.1173 | 0.4762 |
| Eigenvector | GCPI(t) | −0.4506 | −0.1999 | 0.2344 | 0.1349 | 0.7615 | 0.3225 |
| | GIP(t) | 0.4679 | −0.0956 | −0.5554 | 0.4398 | 0.1320 | 0.5026 |
| | GM2(t) | −0.0560 | 0.8195 | −0.3585 | −0.1494 | 0.3862 | −0.1589 |
| | GER(t) | −0.5685 | 0.2274 | −0.1425 | −0.0785 | −0.4770 | 0.6091 |
| Eigenvalues | | 1.4528 | 1.0368 | 0.9829 | 0.9633 | 0.9023 | 0.6619 |
| Proportion | | 0.2421 | 0.1728 | 0.1638 | 0.1606 | 0.1504 | 0.1103 |
| Cumulative Proportion | | 0.2421 | 0.4149 | 0.5788 | 0.7393 | 0.8897 | 1.0000 |

**Table 11.** *Cont.*

| | | | | | | | |
|---|---|---|---|---|---|---|---|
| **Panel D: Russia** | | | | | | | |
| | TURN(t) | 0.4356 | 0.1878 | 0.5487 | −0.3907 | −0.5548 | −0.1157 |
| | PE(t) | 0.5066 | 0.2570 | −0.4655 | 0.3133 | −0.3046 | 0.5194 |
| Eigenvector | GCPI(t) | −0.1440 | 0.7273 | −0.2072 | 0.2147 | −0.0993 | −0.5929 |
| | GIP(t) | 0.4192 | 0.2859 | −0.1720 | −0.5488 | 0.6417 | 0.0028 |
| | GM2(t) | 0.4528 | −0.0407 | 0.4651 | 0.6330 | 0.3890 | −0.1583 |
| | GER(t) | −0.3901 | 0.5351 | 0.4398 | 0.0362 | 0.1627 | 0.5833 |
| Eigenvalues | | 1.5745 | 1.2736 | 0.8889 | 0.8625 | 0.8091 | 0.5914 |
| Proportion | | 0.2624 | 0.2123 | 0.1482 | 0.1437 | 0.1348 | 0.0986 |
| Cumulative Proportion | | 0.2624 | 0.4747 | 0.6228 | 0.7666 | 0.9014 | 1.0000 |
| **Panel E: South Africa** | | | | | | | |
| | TURN(t) | 0.0589 | 0.6031 | 0.5104 | −0.3349 | −0.2354 | −0.4524 |
| | PE(t) | −0.2694 | −0.1360 | 0.6346 | 0.6794 | 0.1858 | −0.1001 |
| Eigenvector | GCPI(t) | −0.1449 | 0.4977 | −0.5682 | 0.5031 | 0.0441 | −0.3918 |
| | GIP(t) | −0.5640 | 0.4681 | 0.0412 | −0.0483 | −0.0769 | 0.6729 |
| | GM2(t) | 0.5067 | 0.3622 | 0.0920 | 0.0240 | 0.7342 | 0.2528 |
| | GER(t) | 0.5727 | 0.1404 | 0.0615 | 0.4125 | −0.6026 | 0.3394 |
| Eigenvalues | | 1.3297 | 1.1928 | 1.0521 | 0.9145 | 0.8336 | 0.6774 |
| Proportion | | 0.2216 | 0.1988 | 0.1753 | 0.1524 | 0.1389 | 0.1129 |
| Cumulative Proportion | | 0.2216 | 0.4204 | 0.5958 | 0.7482 | 0.8871 | 1.0000 |

Note: Principal component analysis results based on $SIX = \alpha + \beta_1 TURN\,(t) + \beta_2 PE\,(t) + \beta_3 GCPI\,(t) + \beta_4 GIP\,(t) + \beta_5 GM2\,(t) + \beta_6 GER\,(t)$.

The above regressions show that the interactions between the sentiment index and various variables tend to be different. For instance, there was a positive relationship between turnover and the sentiment index for all sample markets. This finding signifies that increases in the turnover ratio can create a higher sentiment index in BRICS. Moreover, despite the same correlation between the markets and the same factor, different coefficients denoted different levels of effects on the sentiment index. For example, despite the negative relationship between the growth rate of industrial production and the sentiment index, both in Brazil and South Africa, the element of the industrial production growth ratio in Brazil can exert greater influence on the sentiment index due to the greater absolute value of the coefficient in Brazil compared to that of South Africa. Thus, a 1% rise in the industrial production in Brazil results in an 11.65% fall in the sentiment index, but causes only a 4.82% decline in South Africa.

Granger Causality: *CSAD* and *SIX* (Sentiment Based on Principal Component)

In Table 12 we see the results of tests of Granger causality between sentiment (*SIX*) and *CSAD*. We are mainly interested in the effect of sentiment (*SIX*) on *CSAD*. From the cases of India and China and with reference to pre/during COVID periods (the last two rows), we can see that *SIX* Granger caused *CSAD* (the hypothesis is rejected because the p value is less than 0.10), but this is irrelevant to the presence of COVID or not. In the case of Russia and India, *SIX* Granger caused *CSAD* during the European debt crisis, but this does not seem to be the case for any other country. Overall, *SIX* does not Granger cause *CSAD* as a result of a crisis. There is limited evidence that *SIX* Granger causes *CSAD* and this is NOT the result of a crisis. This means that the null hypothesis (sentiment Granger causes herding or *CSAD*) is partially supported.

**Table 12.** Granger causality: Relationship between *CSAD* and SIX (Sentiment Index).

| Null Hypothesis | Period | Brazil | China | India | Russia | South Africa |
|---|---|---|---|---|---|---|
| **Panel A:** CSAD$_t$ does not Granger Cause SIX$_t$ (Prob.) | Pre-Global Financial Crisis Period | 0.74 | 0.60 | 0.12 | 0.66 | 0.67 |
| | Global Financial Crisis Period | **0.05 \*\*** | 0.97 | 0.17 | 0.48 | 0.24 |
| | Pre-European Debt Crisis Period | **0.04 \*\*** | 0.85 | 0.26 | 0.52 | 0.43 |
| | European Crisis Period | **0.01 \*\*** | 0.30 | **0.04 \*\*** | 0.95 | 0.40 |
| | Pre-COVID-19 Crisis Period | 0.50 | **0.04 \*\*** | 0.90 | 0.17 | 0.90 |
| | COVID-19 CRISIS Period | **0.03 \*\*** | 0.18 | **0.01 \*\*** | 0.65 | **0.01 \*\*** |
| **Panel B:** SIX$_t$ does not Granger Cause CSAD$_t$ (Prob.) | Pre-Global Financial Crisis Period | 0.22 | 0.47 | **0.02 \*\*** | **0.05 \*\*** | 0.28 |
| | Global Financial Crisis Period | 0.97 | 0.50 | 0.14 | 0.98 | 0.83 |
| | Pre-European Debt Crisis Period | 0.43 | 0.15 | 0.69 | 0.51 | 0.10 |
| | European Crisis Period | 0.92 | 0.44 | **0.08 \*\*\*** | **0.00 \*\*** | 0.75 |
| | Pre-COVID-19 Crisis Period | 0.42 | **0.06 \*\*\*** | **0.08 \*\*\*** | 0.17 | **0.00 \*\*** |
| | COVID-19 CRISIS Period | 0.50 | **0.06 \*\*\*** | **0.03 \*\*** | 0.22 | 0.21 |

Note: ** and bold numbers mean significant at 5% level and *** and bold numbers stands for significant at 10% level.

### 4.4. The Effect of the US Stock Market on BRICS

Chow tests are used to detect the presence of structural breaks. Table 13 shows that there are structural breaks at a daily frequency and, for most of the sub-periods, at weekly and monthly frequencies. Table 14 presents results with regard to spillover effects between the US and BRICS considering different time frequencies and crisis events. Spillover effects are captured by a significant and negative $\alpha_5$ in $\left(R_{US,t}{}^2\right)$ in Equation (17) (also presented at the legend of the relevant table). Unequivocally, there are no spillover effects that are fronted by the US to BRICS at lower frequencies (weekly/monthly) regardless of crisis or not. At daily frequencies, the spill-over effect is more pronounced at the Pre-European Debt Crisis, and then the second period during which it is more pronounced is the Global Financial Crisis Period. Generally speaking, spillovers appear to be unrelated to crises or pre-crises periods. When it comes to the intensity of the effect, the spillover effect between the US and India during the Pre-European Debt Crisis is greater than that in other countries (|−4.62| in Brazil, |−5.61| in China, |−12.44| in India, |−3.90| in South Africa), indicating that, despite looking at the same point in time, different markets behave differently. Additionally, Brazil is identified as the market with the strongest spillover effect. It is present in the majority of sub-periods, excluding the COVID-19 Crisis period. The findings here are in agreement with Chiang and Zheng (2010), who remarked that the US market played a significant effect on global financial markets, and the majority of sample markets in their study tended to herd towards the US, including some Latin American markets. Overall, our null hypothesis (US investor behaviour affects investor behaviour in BRICS) is partially supported and it is unrelated to crises.

**Table 13.** Chow Test: Relationship between the US Equity Market and BRICS.

| | | | Brazil | China | India | Russia | South Africa |
|---|---|---|---|---|---|---|---|
| **Panel A: Sub-Period 1 (02/07/2007–31/03/2009)** | | | | | | | |
| (1) Pre-Global Financial Crisis (02/07/2007–14/09/2008) (2) Global Financial Crisis (15/09/2008–31/03/2009) | F-statistic (Prob.) | Daily | **3.84 ** (0.00)** | **1.89 *** (0.09)** | **8.92 ** (0.00)** | **4.48 ** (0.06)** | **5.51 ** (0.00)** |
| | | Weekly | 0.94 (0.45) | 0.75 (0.59) | 1.82 (0.12) | **5.06 (0.00)** | 1.84 (0.11) |
| | | Monthly | 1.50 (0.27) | 0.87 (0.53) | 0.51 (0.76) | 0.77 (0.59) | **3.23 ** (0.04)** |
| **Panel B: Sub-Period 2 (01/04/2009–31/01/2012)** | | | | | | | |
| (1) Pre-European Debt Crisis (01/04/2009–31/03/2010) (2) European Debt Crisis (01/04/2010–31/01/2012) | F-statistic (Prob.) | Daily | **3.79 ** (0.00)** | **3.70 ** (0.00)** | **17.72 ** (0.00)** | **7.49 ** (0.00)** | **31.90 ** (0.00)** |
| | | Weekly | **2.71 ** (0.02)** | **2.91 ** (0.02)** | **3.47 ** (0.01)** | **7.65 ** (0.00)** | **8.85 ** (0.00)** |
| | | Monthly | 2.04 (0.11) | 0.71 (0.62) | 0.83 (0.00) | **2.57 ** (0.05)** | **3.43 ** (0.02)** |
| **Panel C: Sub-Period 3 (01/02/2012–30/09/2021)** | | | | | | | |
| (1) Pre COVID-19 Crisis (01/02/2012–30/12/2019) (2) COVID-19 Crisis (31/12/2019–30/09/2021) | F-statistic (Prob.) | Daily | **29.53 ** (0.00)** | **14.64 ** (0.00)** | **30.42 ** (0.00)** | **53.53 ** (0.00)** | **54.57 ** (0.00)** |
| | | Weekly | **5.55 ** (0.00)** | **3.26 ** (0.01)** | **6.82 ** (0.00)** | **6.39 ** (0.00)** | **20.32 ** (0.00)** |
| | | Monthly | 1.04 (0.40) | **4.43 (0.00)** | **5.25 ** (0.00)** | **2.00 *** (0.08)** | **4.22 ** (0.00)** |

Note: This table shows spillover effect results between US sentiment and BRICS based on the following regression: $CSAD_{i,t} = a_1 + a_2|R_{m,t}| + a_3(R_{m,t})^2 + \alpha_4 CSAD_{us,t} + a_5(R_{us,t})^2 + e_t$. ** and bold numbers mean significant at 5% level and *** and bold numbers indicate significance at 10% level. Global Financial Crisis (15/09/2008–31/03/2009), European Debt Crisis (1/04/2010–31/01/2012) and COVID-19 Crisis (31/12/2019–30/09/2021).

**Table 14.** Spillover Effect(s) between the US Equity Market and BRICS.

| | | | Brazil | China | India | Russia | South Africa |
|---|---|---|---|---|---|---|---|
| **Panel A: Non-Crisis Period** | | | | | | | |
| Pre-Global Financial Crisis (02/07/2007–14/09/2008) | Daily | $R_{(us)\,mt}^2$ (Prob.) | **−5.34 (0.01)** | 1.76 (0.59) | −0.22 (0.94) | −2.06 (0.73) | 1.39 (0.51) |
| | | R-square | 0.31 | 0.02 | 0.28 | 0.69 | 0.50 |
| | Weekly | $R_{(us)\,mt}^2$ (Prob.) | −2.55 (0.42) | 5.03 (0.41) | −7.19 (0.23) | −4.55 (0.21) | −2.05 (0.66) |
| | | R-square | 0.20 | 0.05 | 0.19 | 0.33 | 0.42 |
| | Monthly | $R_{(us)\,mt}^2$ (Prob.) | −1.11 (0.80) | 10.38 (0.31) | −3.12 (0.89) | −3.00 (0.52) | −8.49 (0.16) |
| | | R-square | 0.36 | 0.28 | 0.33 | 0.57 | 0.75 |
| Pre-European Debt Crisis (01/04/2009–31/03/2010) | Daily | $R_{(us)\,mt}^2$ (Prob.) | **−4.62 ** (0.04)** | **−5.61 ** (0.04)** | **−12.44 ** (0.01)** | 30.92 (0.00) | **−3.90 (0.03)** |
| | | R-square | 0.59 | 0.20 | 0.40 | 0.47 | 0.71 |
| | Weekly | $R_{(us)\,mt}^2$ (Prob.) | 3.59 (0.21) | −3.39 (0.35) | **−8.62 ** (0.04)** | **−8.53 *** (0.06)** | −0.01 (0.99) |
| | | R-square | 0.64 | 0.16 | 0.68 | 0.46 | 0.60 |
| | Monthly | $R_{(us)\,mt}^2$ (Prob.) | 9.99 (0.03) | 1.99 (0.70) | −8.41 (0.36) | −0.55 (0.95) | −7.56 (0.14) |
| | | R-square | 0.82 | 0.83 | 0.85 | 0.83 | 0.67 |

**Table 14.** *Cont.*

|  |  |  | Brazil | China | India | Russia | South Africa |
|---|---|---|---|---|---|---|---|
| | | **Panel A: Non-Crisis Period** | | | | | |
| Pre COVID-19 Disease Crisis (01/02/2012–30/12/2019) | Daily | $R_{(us)\,mt}^2$ (Prob.) | **−3.54 **** **(0.01)** | 7.27 (0.00) | −0.17 (0.90) | 5.45 (0.03) | −1.34 (0.36) |
| | | R-square | 0.32 | 0.16 | 0.13 | 0.38 | 0.24 |
| | Weekly | $R_{(us)\,mt}^2$ (Prob.) | 1.11 (0.50) | 2.47 (0.37) | −1.98 (0.32) | 0.19 (0.92) | 1.96 (0.23) |
| | | R-square | 0.38 | 0.24 | 0.09 | 0.22 | 9.20 |
| | Monthly | $R_{(us)\,mt}^2$ (Prob.) | −3.15 (0.14) | 3.15 (0.21) | −1.61 (0.52) | −3.36 (0.20) | −1.83 (0.40) |
| | | R-square | 0.47 | 0.46 | 0.03 | 0.18 | 0.16 |
| | | **Panel B: Crisis Period** | | | | | |
| Global Financial Crisis (15/09/2008–31/03/2009) | Daily | $R_{(us)\,mt}^2$ (Prob.) | **−1.91 ***** **(0.06)** | −0.12 (0.85) | **−1.44 ***** **(0.06)** | −7.30 (0.18) | 0.26 (0.60) |
| | | R-square | 0.60 | 0.15 | 0.51 | 0.47 | 0.73 |
| | Weekly | $R_{(us)\,mt}^2$ (Prob.) | **−3.83 ***** **(0.06)** | −0.29 (0.80) | −2.03 (0.11) | −2.30 (0.17) | −0.80 (0.19) |
| | | R-square | 0.60 | 0.18 | 0.60 | 0.65 | 0.78 |
| | Monthly | $R_{(us)\,mt}^2$ (Prob.) | −9.60 (0.41) | −2.52 (0.47) | **−9.27 **** **(0.07)** | −6.66 (0.52) | 4.31 (0.28) |
| | | R-square | 0.58 | 0.90 | 0.97 | 0.83 | 0.91 |
| European Debt Crisis (01/04/2010–31/01/2012) | Daily | $R_{(us)\,mt}^2$ (Prob.) | 0.53 (0.67) | −0.33 (0.84) | −0.24 (0.82) | 5.16 (0.13) | 0.42 (0.55) |
| | | R-square | 0.40 | 0.09 | 0.15 | 0.40 | 0.56 |
| | Weekly | $R_{(us)\,mt}^2$ (Prob.) | **−2.84 **** **(0.03)** | −1.55 (0.47) | −1.48 (0.44) | 1.41 (0.50) | −2.24 (0.03) |
| | | R-square | 0.51 | 0.41 | 0.15 | 0.28 | 0.56 |
| | Monthly | $R_{(us)\,mt}^2$ (Prob.) | −2.29 (0.22) | −4.56 (0.19) | **−4.05 ***** **(0.08)** | 0.85 (0.81) | 1.33 (0.45) |
| | | R-square | 0.49 | 0.32 | 0.31 | 0.26 | 0.47 |
| COVID-19 Disease Crisis (31/12/2019–30/09/2021) | Daily | $R_{(us)\,mt}^2$ (Prob.) | −0.69 ** (0.41) | −0.28 (0.61) | 0.39 (0.41) | 0.81 (0.05) | −0.94 (0.31) |
| | | R-square | 0.73 | 0.13 | 0.56 | 0.61 | 0.65 |
| | Weekly | $R_{(us)\,mt}^2$ (Prob.) | −1.53 (0.15) | −0.40 (0.65) | −0.89 (0.41) | 0.29 (0.61) | 4.88 (0.00) |
| | | R-square | 0.74 | 0.18 | 0.56 | 0.66 | 0.72 |
| | Monthly | $R_{(us)\,mt}^2$ (Prob.) | 2.97 (0.24) | −2.85 (0.45) | −4.24 (0.28) | −0.48 (0.88) | −6.86 (0.41) |
| | | R-square | 0.76 | 0.17 | 0.58 | 0.46 | 0.82 |

Note: This table illustrates regression results based on the following model $CSAD_{i,t} = a_1 + a_2|R_{m,t}| + a_3(R_{m,t})^2 + \alpha_4 CSAD_{us,t} + a_5(R_{us,t})^2 + e_t$. In addition, different time frequencies have been taken into account to decide whether spillover effect(s) is a short-lived phenomenon or can last for a longer time. ** and bold numbers mean significant at 5% level as well as *** and bold numbers indicate significant at 10% level.

### 4.5. The Effect of US Sentiment/Fear (VIX) on BRICS

Chow test results in Table 15 indicate the presence of structural breaks supporting the breaking up of the total sample in smaller periods. Table 16 presents results of the effect

of the US "fear index" (as captured by the US CBOE implied volatility (*VIX*)) on BRICS, during crisis- and non-crisis periods. Results show that it is mainly Brazil which is affected by *VIX* and this is unrelated to crisis periods or not. Moreover, it is worth noticing (and perhaps this is the greatest finding here) that the effect of *VIX* is greatest during COVID-19 and affects all countries except China. China is the only country which is 'free' of the influence of US sentiment. The reasons for the observed behaviour in China and Brazil are given below. Overall, our null hypothesis ((*VIX*)-fear index affects investor behaviour (causes herding) in BRICS) is partially supported. This is observed only during COVID-19 for all countries except China. The most important reason for the weak connection between the US stock market and the Chinese equity market, is that, technically speaking, the two markets' indexes have a low correlation coefficient. Wroblewska (2016) reported a correlation coefficient of 0.15 between the Shanghai Index and S&P500 in the last two years. More surprising, when extending the length of time to 10 years, Wroblewska (2016) still found that the correlation coefficient was just 0.37. Besides this, it is their different financial systems and regulations. For instance, in order to control volatilities and address risks at a moderate level, a series of regulations have been introduced to the Chinese equity market, such as restrictions on initial public offerings, limits on short sales and daily limits of 10% rises or declines (Wroblewska 2016). However, similar interferences and regulations are not reported in the US.

**Table 15.** Chow Test: US Fear Index or VIX on *CSAD* (Sub-periods).

| | | | Brazil | China | India | Russia | South Africa |
|---|---|---|---|---|---|---|---|
| **Panel A: Sub-Period 1 (02/07/2007–31/03/2009)** | | | | | | | |
| (1) Pre-Global Financial Crisis (02/07/2007–14/09/2008) (2) Global Financial Crisis (15/09/2008–31/03/2009) | F-statistic (Prob.) | Daily | **37.37 \*\*** **(0.00)** | 1.44 (0.22) | **8.92 \*\*** **(0.00)** | **17.35 \*\*** **(0.00)** | **42.75 \*\*** **(0.00)** |
| | | Weekly | **10.25 \*\*** **(0.00)** | 1.11 (0.36) | 0.92 (0.45) | **24.87 \*\*** **(0.00)** | **7.48 \*\*** **(0.00)** |
| | | Monthly | **3.83 \*\*** **(0.03)** | 0.61 (0.62) | 1.14 (0.38) | **8.11 \*\*** **(0.00)** | 0.48 (0.75) |
| **Panel B: Sub-Period 2 (01/04/2009–31/01/2012)** | | | | | | | |
| (1) Pre European Debt Crisis (01/04/2009–31/03/2010) (2) European Debt Crisis (01/04/2010–31/01/2012) | F-statistic (Prob.) | Daily | **9.72 \*\*** **(0.00)** | **10.38 \*\*** **(0.00)** | **40.85 \*\*** **(0.00)** | **15.25 \*\*** **(0.00)** | **85.35 \*\*** **(0.00)** |
| | | Weekly | **5.58 \*\*** **(0.00)** | **3.91 \*\*** **(0.00)** | **8.13 \*\*** **(0.00)** | **12.53 \*\*** **(0.00)** | **19.63 \*\*** **(0.00)** |
| | | Monthly | 0.42 (0.79) | 0.95 (0.45) | **3.55 \*\*** **(0.02)** | 1.99 (0.13) | **6.36 \*\*** **(0.00)** |
| **Panel C: Sub-Period 3 (01/02/2012–30/09/2021)** | | | | | | | |
| (1) Pre COVID-19 Disease Crisis (01/02/2012–30/12/2019) (2) COVID-19 Disease Crisis (31/12/2019–30/09/2021) | F-statistic (Prob.) | Daily | **33.92 \*\*** **(0.00)** | 15.22 (0.00) | **78.39 \*\*** **(0.00)** | **68.74 \*\*** **(0.00)** | **188.81 \*\*** **(0.00)** |
| | | Weekly | **3.74 \*\*** **(0.01)** | **3.61 \*\*** **(0.01)** | **12.82 \*\*** **(0.00)** | **2.60 \*\*** **(0.04)** | **43.s38 \*\*** **(0.00)** |
| | | Monthly | 0.57 (0.69) | **7.37 \*\*** **(0.00)** | **5.85 \*\*** **(0.00)** | 0.67 (0.62) | **8.03 \*\*** **(0.00)** |

Note: This table shows spillover effect results between US sentiment and BRICS based on the following model $CSAD_{i,t} = a_1 + a_2|R_{m,t}| + a_3(R_{m,t})^2 + a_4 VIX_{US,t} + e_t$. The US sentiment index is captured by VIX. \*\* and bold numbers mean significant at 5% level. Every sub-period covers one crisis, which are Global Financial Crisis (15/09/2008–31/03/2009), European Debt Crisis (1/04/2010–31/01/2012) and COVID-19 Crisis (31/12/2019–30/09/2021).

**Table 16.** The effect of US 'fear index' or VIX on *CSAD* in BRICS.

| | | | Brazil | China | India | Russia | South Africa |
|---|---|---|---|---|---|---|---|
| | | **Panel A: Non-Crisis Period** | | | | | |
| Pre-Global Financial Crisis (02/07/2007–14/09/2008) | Daily | VIX $_{US (t)}$ (Prob.) | **−0.0054 \*\*\*** **(0.06)** | 0.0018 (0.71) | −0.0042 (0.36) | −0.0074 (0.41) | −0.0022 (0.51) |
| | | R-squared | 0.23 | 0.02 | 0.23 | 0.67 | 0.42 |
| | Weekly | VIX $_{US (t)}$ (Prob.) | **−0.02 \*\*** **(0.03)** | −0.01 (0.35) | −0.0068 (0.62) | −0.0017 (0.83) | **−0.02 \*\*** **(0.05)** |
| | | R-squared | 0.10 | 0.05 | 0.02 | 0.31 | 0.38 |
| | Monthly | VIX $_{US (t)}$ (Prob.) | −0.0070 (0.73) | 0.0015 (0.98) | 0.07 (0.22) | −0.02 (0.48) | −0.02 (0.70) |
| | | R-squared | 0.26 | 0.04 | 0.41 | 0.49 | 0.32 |
| Pre-European Debt Crisis (01/04/2009–31/03/2010) | Daily | VIX $_{US (t)}$ (Prob.) | **−0.0074 \*\*** **(0.05)** | 0.0010 (0.95) | −0.0012 (0.89) | 0.02 (0.26) | −0.01 (0.14) |
| | | R-squared | 0.42 | 0.10 | 0.23 | 0.38 | 0.54 |
| | Weekly | VIX $_{US (t)}$ (Prob.) | 0.0016 (0.84) | 0.03 (0.02) | 0.0055 (0.72) | −0.02 (0.24) | 0.0021 (0.80) |
| | | R-squared | 0.46 | 0.18) | 0.47 | 0.10 | 0.20 |
| | Monthly | VIX $_{US (t)}$ (Prob.) | 0.01 (0.69) | −0.03 (0.23) | 0.08 (0.04) | 0.06 (0.41) | −0.05 (0.22) |
| | | R-squared | 0.58 | 0.84 | 0.90 | 0.51 | 0.29 |
| Pre COVID−19 Crisis (01/02/2012–30/12/2019) | Daily | VIX $_{US (t)}$ (Prob.) | −0.0022 (0.05) | 0.0004 (0.82) | 0.0019 (0.06) | 0.0044 (0.02) | −0.0006 (0.57) |
| | | R-squared | 0.29 | 0.16 | 0.13 | 0.38 | 0.18 |
| | Weekly | VIX $_{US (t)}$ (Prob.) | −0.0011 (0.66) | 0.0038 (0.35) | 0.0037 (0.21) | −0.0006 (0.83) | −0.0014 (0.55) |
| | | R-squared | 0.32 | 0.24 | 0.09 | 0.21 | 0.15 |
| | Monthly | VIX $_{US (t)}$ (Prob.) | −0.0001 (0.98) | 0.01 (0.49) | −0.0004 (0.96) | 0.0023 (0.77) | 0.0034 (0.61) |
| | | R-squared | 0.33 | 0.45 | 0.02 | 0.15 | 0.04 |
| | | **Panel B: Crisis Period** | | | | | |
| Global Financial Crisis (15/09/2008 -31/03/2009) | Daily | VIX $_{US (t)}$ (Prob.) | −0.0034 (0.58) | 0.0027 (0.61) | **−0.02 \*\*** **(0.01)** | −0.0050 (0.90) | −0.0041 (0.31) |
| | | R-squared | 0.55 | 0.12 | 0.47 | 0.46 | 0.70 |
| | Weekly | VIX $_{US (t)}$ (Prob.) | −0.01 (0.55) | 0.03 (0.16) | **−0.03 \*\*\*** **(0.08)** | −0.02 (0.42) | 0.01 (0.44) |
| | | R-squared | 0.46 | 0.17 | 0.52 | 0.60 | 0.70 |
| | Monthly | VIX $_{US (t)}$ (Prob.) | 0.08 (0.28) | 0.0047 (0.78) | −0.0021 (0.96) | −0.10 (0.30) | −0.0066 (0.86) |
| | | R-squared | 0.55 | 0.85 | 0.77 | 0.74 | 0.82 |
| European Debt Crisis (01/04/2010 -31/01/2012) | Daily | VIX $_{US (t)}$ (Prob.) | 0.0009 (0.56) | 0.0003 (0.89) | −0.0002 (0.89) | 0.02 (0.00) | −0.0006 (0.56) |
| | | R-squared | 0.32 | 0.09 | 0.15 | 0.36 | 0.65 |
| | Weekly | VIX $_{US (t)}$ (Prob.) | −0.0005 (0.87) | 0.01 (0.16) | 0.0029 (0.53) | −0.0073 (0.17) | −0.0012 (0.64) |
| | | R-squared | 0.36 | 0.42 | 0.14 | 0.27 | 0.48 |
| | Monthly | VIX $_{US (t)}$ (Prob.) | 0.02 (0.15) | 0.0048 (0.78) | 0.0025 (0.83) | 0.0047 (0.76) | 0.0032 (0.59) |
| | | R-squared | 0.40 | 0.85 | 0.15 | 0.26 | 0.37 |

**Table 16.** *Cont.*

|  |  |  | Brazil | China | India | Russia | South Africa |
|---|---|---|---|---|---|---|---|
|  |  | **Panel B: Crisis Period** | | | | | |
| **COVID-19 Crisis (31/12/2019 -30/09/2021)** | Daily | VIX $_{US\,(t)}$ (Prob.) | **−0.0121 ** ** (0.00)** | −0.0010 (0.69) | **−0.0040 *** (0.09)** | **−0.0050 ** (0.02)** | **−0.0142 ** (0.00)** |
|  |  | R-squared | 0.55 | 0.12 | 0.49 | 0.52 | 0.40 |
|  | Weekly | VIX $_{US\,(t)}$ (Prob.) | **−0.0131 ** (0.04)** | −0.0061 (0.31) | −0.0078 (0.24) | −0.0003 (0.95) | −0.02 (0.11) |
|  |  | R-squared | 0.62 | 0.13 | 0.25 | 0.51 | 0.33 |
|  | Monthly | VIX $_{US\,(t)}$ (Prob.) | **−0.0195 ** (0.05)** | 0.0077 (0.66) | 0.0067 (0.67) | 0.0052 (0.69) | 0.0091 (0.61) |
|  |  | R-squared | 0.77 | 0.04 | 0.52 | 0.29 | 0.75 |

Note: This table shows the effect results of US sentiment on BRICS based on the following model $CSAD_{i,t} = a_1 + a_2|R_{m,t}| + a_3(R_{m,t})^2 + a_4VIX_{US,t} + e_t$. The US sentiment index is captured by VIX. Moreover, crisis effect(s) has also been considered. ** and bold numbers mean significant at 5% level. Similarly, *** and bold numbers mean significant at 10% level.

Brazil is recognised as a country which is influenced the most by US sentiment even in non-crisis periods. This finding goes hand in hand with the findings shown in Table 14, which can confirm the close connection between the US and Brazil. Possible reasons are: (i) that the US and Brazil have established a partnership and strengthened cooperation covering various aspects (such as: security, economy and politics) in order to promote sustainable economic growth, (ii) the US, as the second largest trading partner, has brought lots of trade and contracts in goods in the last two decades rising from $28.2 million in 2002, to $60.7 million in 2008, achieving the peak of $104.3 billion in 2009 (Bodman et al. 2011; U.S. Department of State 2021), (iii) despite experiencing declines at the beginning of 2021, foreign investment flows returned to the Brazilian stock markets from the second quarter, achieving more than $44 million between April and June (Feliba and Lozano 2021), (iv) the US also provided support during COVID-19 to the tune of $16.9 million from government funding and $75 million from the U.S. private sector (U.S. Department of State 2021).

Now when it comes to the effect of US sentiment/fear considering individual crises, Table 16 shows that the US sentiment/fear had a greater effect on BRICS (excluding China), during the COVID-19 period. US sentiment/fear had no effect during the other two crises (Global financial and European sovereign debt crises). The main reason for the greater effect of COVID-19 is that the unpredictable outbreak of COVID-19 is not only a threat to people's health, but also exerts adverse influence on the majority of stock markets all around the world due to a rise in panic and fear sentiment. This kind of sentiment spreads to other markets (Liu et al. 2020). Secondly, in order to slow the rate of infection, companies have to reduce activities and cut down on their labour force, especially in labour-intensive and manufacturing industries, resulting in panic (Liu et al. 2020). The decrease of economic activity makes investors lose confidence in companies' future profitability, and future global economic development, which is reflected in negative equity returns and a decline in stock prices (Liu et al. 2020). Moreover, the lack of confidence in equity markets causes convergence in trading behaviour. Furthermore, as mentioned previously, herding is a short-lived phenomenon and, as can be seen in Table 16, the effect is more prevalent when using daily data.

*4.6. Granger Causality:* CSAD *and VIX*

Table 17 presents Granger causality results. According to Table 17, any causality between *VIX* and *CSAD* (if present in either direction) is not affected by crises. To clarify, causalities are independent of crises. Considering results for individual countries, the country which presents the greatest number of causalities in both directions is Brazil. It seems that there is a two-way causality between *VIX* and *CSAD*, especially before but also during the COVID period. This result is not surprising given the relationship between

Brazil and the US discussed in the previous section. A similar situation is observed in Russia. Causality runs in both directions, especially during the COVID period. This is consistent with results presented in Table 16. As for the reason(s) why the Russian market is so closely linked to the US market, this can be explained by an increase in Foreign Capital Flows into Russia. For instance, there is more than 50% of foreign capital inflows that can be ascribed to North American investors compared to European investors (just 26% for European accounts) (Rapoza 2019). Simultaneously, a report by the Moscow Exchange pointed out that US fund managers increased their investment by 58% into Russian publicly listed corporations during the second quarter of 2019 compared to that of 2015, and reached $79.3 billion (Rapoza 2019). Overall causality between *CSAD* and VIX if present are independent of crises and run in both directions, while previously it was thought that there is a causality running from VIX to CSAD.

**Table 17.** Granger causality tests: *CSAD* and VIX—Daily Data.

| Null Hypothesis | Period | Brazil | China | India | Russia | South Africa |
|---|---|---|---|---|---|---|
| **Panel A:** $CSAD_{i,t}$ does not cause $VIX_{U.S}$(Prob.) | Pre-Global Crisis Period | 0.39 | **0.07 *** | 0.62 | 0.88 | 0.54 |
| | Global Crisis Period | 0.72 | 0.46 | 0.34 | 0.68 | 0.41 |
| | Pre-European Debt Crisis Period | 0.49 | 0.28 | 0.73 | 0.44 | 0.66 |
| | European Crisis Period | **0.09 *** | 0.64 | 0.48 | 0.16 | **0.02 ** |
| | Pre-COVID-19 Crisis Period | **0.02 ** | **0.08 *** | 0.96 | 0.11 | 0.59 |
| | COVID-19 CRISIS Period | **0.01 ** | 0.84 | 0.52 | **0.01 ** | 0.54 |
| **Panel B:** $VIX_{U.S(t)}$ does not cause $CSAD_{i,t}$(Prob.) | Pre-Global Crisis Period | 0.28 | 0.36 | 0.33 | 0.27 | 0.62 |
| | Global Crisis Period | 0.51 | **0.03 ** | 0.21 | **0.01 ** | 0.44 |
| | Pre-European Debt Crisis Period | **0.03 ** | 0.56 | 0.95 | 0.15 | 0.58 |
| | European Crisis Period | **0.08 *** | 0.27 | 0.91 | 0.14 | **0.00 ** |
| | Pre-COVID-19 Crisis Period | **0.03 ** | **0.01 ** | **0.10 *** | 0.14 | **0.00 ** |
| | COVID-19 CRISIS Period | **0.07 *** | 0.56 | 0.46 | **0.01 ** | 0.36 |

Note: ** and bold numbers mean significant at 5% level and *** and bold numbers stands for significant at 10% level.

## 5. Conclusions

This study set out to investigate determinants of investors' herding in BRICS. The purpose of this project was to detect whether investors displayed herding when they were placed in the same period or faced the same events.

In the first instance, this study focused on the effects of crises on investors' behaviour. Multiple sub-periods were considered, in order to determine whether investors' decisions were influenced by varying degrees of market stress and if they changed their behaviour patterns during crises. The global financial crisis, the European sovereign debt crisis and COVID were considered. The whole sample is from 2-7-07 until 30-9-21. One of the innovations of this study is that it incorporates all three most recent crises (considering pre-crisis and during-crisis sub-samples correcting for structural breaks) and looks at BRICS behaviour simultaneously. Most research on BRICS has concentrated on individual countries (for example, Demirer and Kutan (2006) and Tan et al. (2008) on the Chinese market, Lakshman et al. (2013) and Banerjee and Padhan (2017) on the Indian market, Ababio and Mwamba (2017) on South Africa, Indārs et al. (2019) on Russia, and Júnior et al. (2019) on Brazil). The most recent study, that was brought to our attention at the final stages of this study and looks into herding in BRICS simultaneously, is that of Mulki and Rizkianto (2020). Mulki and Rizkianto (2020) are not looking exactly into the same period, and they are not considering pre- and during crisis behaviour, appear to be correcting for possible

structural breaks in their sample, or consider different time frequencies as we do. Despite this, we managed to identify two common research elements between the two studies (such as herding and the effect of volatility on herding) which were previously discussed in the 'contribution section' and 'BRICS Research section', indicating how our study differs and innovates. To summarise, (and running the risk of becoming repetitive) both studies agree on findings regarding India (no herding), Russia (no herding) and China (herding), but there is no common ground as to the effect of volatility (high/low) on herding.

Additionally, time-frequency was taken into account in order to differentiate between short-lived and long-lived herding phenomena. Herding is a short-term phenomenon. Herding is more likely to emerge when daily data is utilised, but disappears at monthly intervals. Herding is independent of crises and limited. We observe little or no herding before any of the crises or during the crises considered here. At this point, it is worth noting that the Global Financial crisis seemed to have the greatest effect compared to non-financial crises (such as COVID-19). Most importantly, China was the only country to show herding, mainly during the Global Financial Crisis, using daily data. No significant differences in investors' behaviour were observed during crisis and non-crisis periods for the other four sample markets. Once more, our hypothesis that crises have an impact on herding is not supported (see Table 4).

Next, we investigated the effect of *CSAD* on volatility. The effect of herding on volatility is an open issue (see Blasco et al. (2012) and Alemanni and Ornelas (2006)). Research has produced conflicting results and is not considering multiple crises periods as we do. The effect of *CSAD* on volatility is positive (Table 7). A low *CSAD* means that investors tend to behave in a similar way, implying herding. A positive relationship between volatility and *CSAD* indicates that a decrease in the *CSAD* (convergence in opinions) results in a decline in volatility (the lower the CSAD, the lower the volatility), and vice versa. An increase in *CSAD* means that "individual returns deviate from the market return" (Chang et al. 2000), this in turn means diversity in opinions which results in greater volatility. The null hypothesis (a low *CSAD* causes an increase in volatility) is not supported.

Additionally, this study was designed to assess the effect of volatility on herding. Six different volatility measures were used for a more comprehensive analysis of the effects of volatility on investor behaviour. We have not come across another study that employs six different volatility measures to capture the effect of volatility on herding for all BRICS under different market conditions, so we consider this an important innovation. Most studies employ a single measure (see Huang et al. (2015), Huang and Wang (2017) and Mulki and Rizkianto (2020)). After eliminating the influence of 'volume' and 'day-of-the-week' effects, final volatilities (values) were obtained, and a quartile approach was introduced to examine how the degree of volatility (high/medium/low) affected herding. Multiple regression analysis revealed that all volatility measures (the only exception is the capitalisation-weighted monthly historical volatility measure) have an impact on the CSAD, but only for the high volatility period. We observe significant negative coefficients only in Group/period 4. This is the highest volatility period/group. Specifically, in Brazil, China and India, volatility measures affect CSAD, but only for the most volatile period, while in Russia and South Africa, volatility measures have no effect on herding regardless of volatility intensity (high/low). Given the above, one can safely conclude that volatility affects *CSAD* only in turbulent periods and it is measure dependent. This last finding is specific only in this study since it is the only one that uses six different volatility measures and can actually test for volatility measure homogeneity. Another important finding is that the effect of volatility on herding/CSAD in China tends to be stronger than that in India and in Brazil, as shown by the greater coefficient for all volatility measures used, such as, |−9.37| in China vs. |−1.11| in India when using GARCH, and |−8.55| in China vs. |−0.89| in Brazil when using GK. Most importantly, and similar to previous studies (such as: Lakshman et al. (2013)), high volatility is one of the most important determinants of herding due to rises in uncertainty and fear of losses, resulting in a convergence in investors' behaviour and mimicking.

Thirdly, this study showed that there is limited evidence (some might argue that it verges on NO evidence) that there is Granger causality between the *SIX* (sentiment index based on principal component analysis) and CSAD. Researchers such as Chen et al. (2014), Liao et al. (2011), Hudson (2014) and He et al. (2017) have used principal component analysis to capture sentiment, but they did not report its relationship to herding (unidirectional or bidirectional) under different market conditions and crises which we believe is another innovation. Specifically, there was almost no relationship between the SIX and *CSAD* during the Pre-Global Financial Crisis and Global Financial Crisis, while for the other sub-periods, partial or one-way causality between investors' sentiment and CSAD/herding effect can be reported. From the cases of India and China, and with reference to pre/during COVID periods (the last two rows), we can see that the SIX Granger causes *CSAD* (the hypothesis is rejected because then $p$ value is less than 0.10), but this is irrelevant to the presence of COVID or not. In the case of Russia and India, the SIX Granger caused *CSAD* during the European debt crisis, but this does not seem to be the case for any other country. Overall, the SIX does not Granger cause *CSAD* as a result of a crisis. There is limited evidence that the SIX Granger causes *CSAD* and this is NOT the result of a crisis. This means that the null hypothesis (sentiment Granger causes CSAD/herding) is partially supported. The SIX does not have the effect that we previously thought that it had.

This study also examined spill-over effects between the US equity market and BRICS (Table 14). The US market emerged in the past as another factor that causes herding in BRICS. Analysis showed that all markets showed herding patterns to some extent, using different time frequencies and different sub-periods, but the degree of influence varied from market to market. Spillover effects between the US and BRICS were short-lived. In addition, US sentiment (captured by the *VIX*, an indicator of "fear", Table 16) exerted a great effect on behaviour in four of our sample markets—Brazil, India, Russia and South Africa. The analysis showed that China is the only country which was free of the effects of US "fear", regardless of any crisis or non-crisis period. Conversely, Brazil was the most sensitive market to US sentiment (an explanation is provided in the analysis section for both China and Brazil). Given those findings and the absence of any work on causality between the *VIX* and CSAD/herding for different periods/crises (research in the area employs mostly multivariate regressions and does not consider causalities, see, for example, Bathia et al. (2016) and Economou et al. (2018)), the next step was to run Granger causality tests. Granger causality tests between *CSAD* and the VIX (Table 17) reported no Granger causality for the majority of the periods considered. There are variations between countries when considering sub-periods. The overall causality between *CSAD* and the VIX, if present, are independent of crises and run in both directions, while previously it was thought that causality runs from the *VIX* to CSAD. This finding is unique since it is the only study that investigated causality between the *VIX* and *CSAD* for different periods/crises.

Future research in BRICS could concentrate on more recent crises and their impact. We have considered financial crises as well as non-financial crises (COVID). Perhaps it would make sense to examine the effect of political/military crises. Currently, as we are working on this article, Russia is invading the Ukraine and financial markets are tumbling down uncontrollably, while the price for oil and natural gas is going through the roof. Considering that Russia is part of BRICS, perhaps it would be worth investigating how this affects all BRICS using high frequency data. Given that the UK, US and European union have already announced sanctions against Russia, it would be worth investigating the reaction of all those markets, spillovers and, of course, causalities in a panel framework along the lines of this study.

## 6. Implications

A major implication of this study is that investors should not worry too much about herding when they think about their international investment strategy unless they are investing in China. Herding appears to be present only in China and it is short-lived (H2). Furthermore, herding is unrelated to crises (H1). If herding is perceived as a market

anomaly and they plan to invest in China (which is quite common nowadays given the huge development in this country) then they should not be surprised if observed market movements do not necessarily follow their expectations (due to herding). Having said that, at this point we also need to stress that China is the only country within BRICS which is not affected by the VIX, therefore, China will be a 'counterbalancing force' to the effect(s) of the VIX in an international portfolio (H8 and H9).

Now, with regards to the relationship between volatility and herding, this study shows that (increased) herding has no effect on volatility (H3), but a high volatility causes more herding (H4). This means that investors should be particularly wary when investing in BRICS at periods of high volatility. If their initial investment strategy was to invest only in BRICS in an attempt to achieve a greater return for their portfolio, in periods of greater volatility they should consider investment in countries which correlate negatively to BRICS, or if they insist on investing solely on BRICS, they should consider rebalancing their portfolios, increasing portfolio weights towards Russia and South Africa which appear to be immune to the effect of volatility on herding. The effect of volatility on herding also has implications for the local financial authorities/regulators. Given the findings of this study, it is a dead certainty that high volatility will cause herding; therefore, local regulators should introduce measures to reduce volatility so as to avoid unwanted swings. Finally, investors, before making decisions, should consider how they measure volatility because, as this study has shown, there is no such thing as volatility measure homogeneity (H5) and different measures could lead to different decisions which are not always appropriate.

Regarding the effect of sentiment and US investment behaviour on herding (H6 and H7), there is limited evidence to suggest that their role is significant. What emerges very clearly form this study is that any herding effects observed are unrelated to crises and they are country specific. Therefore, it is a bit difficult to devise an investment strategy that will 'fit' all BRICS countries based on sentiment and US investment behaviour.

Finally, the VIX, which has been considered the greatest predictor in the world of finance (particularly for herding as discussed in the literature), has turned out to be a good predictor only during COVID-19, otherwise its effect on herding is at best limited (H8). At this point it is worth reminding the reader that China is immune to VIX effect(s), in sharp contrast to all other counties. This is why we called it a 'counterbalancing force' at the beginning of this section. In addition, Granger causality indicates a two-way relationship between the VIX and herding (H9), implying that the all-powerful *VIX* is not what most financial economist believe. The findings and implications of this study are relevant to investment theory, behavioural finance and particularly to international investors who wish to diversify their portfolios and invest in BRICS, taking advantage of the opportunities offered there, keeping their risk to a minimum while maximising their returns.

**Author Contributions:** Editing, reviewing, supervision, corrections, administration is E.G., the supervisor of H.Z., and the rest is H.Z. All authors have read and agreed to the published version of the manuscript.

**Funding:** This research received no external funding.

**Institutional Review Board Statement:** Not applicable.

**Informed Consent Statement:** Not applicable.

**Data Availability Statement:** All data is from DataSt ream and Bloomberg.

**Conflicts of Interest:** The authors declare no conflict of interest.

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
