# Peer review of "Measures of Volatility, Crises, Sentiment and the Role of U.S. ‘Fear’ Index (VIX) on Herding in BRICS (2007–2021)"

_jrfm, doi:10.3390/jrfm15030134_

Round 1
Reviewer 1 Report
This is a good paper that fits very well JRFM. I found it both relevant and well written. I have though a few suggestions:
1) Expand the sample to consider a more consistent sample before the financial crisis from 2008-2009.
2) Test for the equality of coefficients over different samples.
3) Consider a time-varying perspective.
Author Response
MEASURES OF VOLATILITY, CRISES, SENTIMENT AND THE ROLE OF U.S. ‘FEAR’ INDEX (VIX) ON HERDING IN BRICS (2007-2021).
We received comments from three different referees, but we do not really know who is the first, second or third referee so we call you randomly referee 1, referee 2 and referee 3.
This is a good paper that fits very well JRFM. I found it both relevant and well written. I have though a few suggestions: (thank you)
Comment 1) Expand the sample to consider a more consistent sample before the financial crisis from 2008-2009.
Authors reply: We have expanded the sample in both directions. The sample now starts in 2007 and ends in Sep 2021. We have also split the whole sample before-the-crisis and during-the-crisis. Three crises are considered, the global financial crisis, the European sovereign debt crisis and covid 19. Expanding before 2007 is not really an option for us because there is lots of data missing which means that our sample will be unbalanced.
Comment 2) Test for the equality of coefficients over different samples.
Authors reply: Even though we were not sure what this comment meant at the beginning, when we asked for clarifications, we were told to check for structural breaks using the Chow test. Now before each regression we run Chow tests. Chow tests support our decision to break the original sample in 3 different sub-samples, before and during each crisis. the time series for all variables is stationary.
Comment 3) Consider a time-varying perspective.
Authors reply: We have run the analysis on a daily, weekly and monthly basis for all regressions and all models. Results indicate that herding is prevalent when daily data is used. This is one of our hypotheses. Different frequencies do not seem to make much difference for all the other hypotheses considered.
Our changes within the document are in green. This will help you spot the changes even though the whole document has changed considerably and most of it is green now. We did not change the color of the tables, but the changes are evident and tables which report Chow results start with ‘Chow test’.
Thank you and we are terribly sorry for the delay but lots of things went wrong (actually anything that could go wrong, did go wrong). Also expanding, splitting the sample and using different frequencies meant that the number of regressions that we had to run multiplied. This also contributed to the delay.
We look forward to hearing from you.
Hang and Evangelos

Reviewer 2 Report
I thank the editor and the author to give me the opportunity to read this paper. I think that the paper deals with a relevant and original topic. The contribution is original and relevant. I just have some comments and recommendations to improve the paper:
- In the introduction, the purpose of the paper could be explained in a clearer way. In the introduction, the authors refer several times to the footnotes. In my opinion, the footnotes should be placed within the text to better justify both the purpose and the literature gap. Once the gap and the purpose have been defined in the introduction, there is no need to remark it to avoid repetition in the text.
- The literature section should be extended and structured better. I really suggest explicit hypotheses. The paper is quantitative and asks various research questions, the suggestion is to structure the literature in terms of hypotheses. Furthermore, some aspects still remain to be clarified and deepened in literature, especially literature concerning the crisis and the relationship between the crisis and the other dimensions. Moreover, the discussion should be revised as a hypothesis development discussion.
- In the section Discussion and Findings, it is useful to put the tables inside the section and not as an appendix. As specified in the previous point, the discussion must be structured considering the hypotheses.
- In the conclusion, the implications should be reported and discussed. It would be interesting to have a specific section for implications (to theory and practice). In the theoretical implications, you should emphasize more to which streams you are contributing to.
Good luck
Author Response
MEASURES OF VOLATILITY, CRISES, SENTIMENT AND THE ROLE OF U.S. ‘FEAR’ INDEX (VIX) ON HERDING IN BRICS (2007-2021).
We received comments from three different referees, but we do not really know who is the first, second or third referee so we call you randomly referee 1, referee 2 and referee 3.
I thank the editor and the author to give me the opportunity to read this paper. I think that the paper deals with a relevant and original topic. The contribution is original and relevant. I just have some comments and recommendations to improve the paper: (thank you)
- Comment 1: In the introduction, the purpose of the paper could be explained in a clearer way. In the introduction, the authors refer several times to the footnotes. In my opinion, the footnotes should be placed within the text to better justify both the purpose and the literature gap. Once the gap and the purpose have been defined in the introduction, there is no need to remark it to avoid repetition in the text.
- Authors reply: Thank you very much for your suggestion. The introduction has changed a lot as you can see. Hopefully it reads better this time. we have moved all footnotes within the text. As you can see the whole document has changed a lot. all new changes are in green. The hypotheses are in red so that they are easier to spot.
- Comment 2: The literature section should be extended and structured better. I really suggest explicit hypotheses. The paper is quantitative and asks various research questions, the suggestion is to structure the literature in terms of hypotheses. Furthermore, some aspects still remain to be clarified and deepened in literature, especially literature concerning the crisis and the relationship between the crisis and the other dimensions. Moreover, the discussion should be revised as a hypothesis development discussion.
Authors reply: We have added our research hypotheses within the literature section (also in introduction) and at the end of the literature review as well. Now we connect the literature review to our hypotheses and we even state in brackets if the hypothesis is supported (or not) in this paper. we think that the reader will find this more informative. Also, at the end of the literature review, we state our research hypotheses again and include the number of tables which deals with each hypothesis. In this way, each hypothesis is connected to a specific table and a specific result. We also ‘connect’ the number of each table to each hypothesis in the conclusion. The approach we have used is ‘hypothesis statement-number of table where each hypothesis is discussed and result’. We hope that this approach will make things clearer for everybody.
- Comment 3: In the section Discussion and Findings, it is useful to put the tables inside the section and not as an appendix. As specified in the previous point, the discussion must be structured considering the hypotheses.
- Authors reply: Our intention is to move the tables within discussion/findings after acceptance. Tables ‘move around’ a lot when we add paragraphs, and this is something that the production team of the journal works on. Sometimes they move the tables in different pages to match their requirements. For the time being we have kept the tables at the end, but we will move those following instructions from the production team. Some of the tables perhaps should be in separate pages or in landscape mode. This is the reason we have kept all tables at the end for the time being.
- Comment 4: In the conclusion, the implications should be reported and discussed. It would be interesting to have a specific section for implications (to theory and practice). In the theoretical implications, you should emphasize more to which streams you are contributing to.
- Authors reply: We have added a new section for this. we have also connected this section to our hypotheses.
- Good luck (Thank you)
Thank you and we are terribly sorry for the delay but lots of things went wrong (actually anything that could go wrong, did go wrong). Also expanding, splitting the sample and using different frequencies meant that the number of regressions that we had to run multiplied. This also contributed to the delay.
We look forward to hearing from you.
Hang and Evangelos

Reviewer 3 Report
Below you can find some comments regarding your paper:
- It is necessary to format your manuscript according to the journal guidelines; Eliminate the footnotes and introduce the relevant comments into the main text;
- Please include in the chapter on the literature review links with the previous research of the authors on this subject, as well as in the chapter on the results and discussions, in order to highlight the improvement of the studies;
- Please check your English spelling, punctuation and word choice for better understanding;
- It is mandatory to mention the exact source for your data and include it in the reference list.
Author Response
MEASURES OF VOLATILITY, CRISES, SENTIMENT AND THE ROLE OF U.S. ‘FEAR’ INDEX (VIX) ON HERDING IN BRICS (2007-2021).
We received comments from three different referees, but we do not really know who is the first, second or third referee so we call you randomly referee 1, referee 2 and referee 3.
Comment 1) It is necessary to format your manuscript according to the journal guidelines; Eliminate the footnotes and introduce the relevant comments into the main text;
Authors reply: done and thank you
Comment 2) Please include in the chapter on the literature review links with the previous research of the authors on this subject, as well as in the chapter on the results and discussions, in order to highlight the improvement of the studies.
Authors reply: we have done this and also added hypotheses as other reviewers have suggested.
Comment 3) Please check your English spelling, punctuation and word choice for better understanding.
Authors reply: we always ask independent editors to check our English before submission to a journal. If there are any specific sections that you would like us to have another look please let us know.
Comment 4) It is mandatory to mention the exact source for your data and include it in the reference list.
Authors reply: All data is from DataStream and Bloomberg. We have added this at the very begining of the document and in the data section. These are databases that our university subscribes to.

Round 2
Reviewer 1 Report
I found that the authors have adequately addressed the issues raised by the reviewers. The paper looks very nice. I recommend its acceptance.
Author Response
THANK YOU SO MUCH FOR ACCEPTING THE MANUSCRIPT,
BEST
HANG AND EVANGELOS
Reviewer 2 Report
In my opinion, the conclusions need to be improved. The conclusions reported in the paper are a repetition of the discussion. In a scientific work the conclusions must reflect the contributions compared to other papers, and must also give opportunities for future work.
With the best.
Reviewer 3 Report
Below you can find some comments regarding your paper:
1. You mention in contribution section: „However, these studies have just investigated herding behaviour in individual markets rather than all of BRICS”.
Still, there are research that investigate herding in BRICS, using the same Chang methods (CSAD): ( such as https://www.researchgate.net/publication/339998992_Herding_Behavior_in_BRICS_Countries_during_Asian_and_Global_Financial_Crisis), which are not mentioned in your references.
Please refere to these studies and compare results, mentioning which is the novelty of your research.
